# Contacts at the Nanoscale and for Nanomaterials

**DOI:** 10.3390/nano14040386

**Published:** 2024-02-19

**Authors:** Hei Wong, Jieqiong Zhang, Jun Liu

**Affiliations:** 1Department of Electrical Engineering, City University of Hong Kong, Hong Kong, China; 2Hubei Jiu Feng Shan Laboratory, Wuhan 430074, China; zhangjieqiong@jfslab.com.cn (J.Z.); liujun@jfslab.com.cn (J.L.)

**Keywords:** metal/semiconductor contact, Schottky contact, ohmic contact, 2D material/metal contacts, contact scaling

## Abstract

Contact scaling is a major challenge in nano complementary metal–oxide–semiconductor (CMOS) technology, as the surface roughness, contact size, film thicknesses, and undoped substrate become more problematic as the technology shrinks to the nanometer range. These factors increase the contact resistance and the nonlinearity of the current–voltage characteristics, which could limit the benefits of the further downsizing of CMOS devices. This review discusses issues related to the contact size reduction of nano CMOS technology and the validity of the Schottky junction model at the nanoscale. The difficulties, such as the limited doping level and choices of metal for band alignment, Fermi-level pinning, and van der Waals gap, in achieving transparent ohmic contacts with emerging two-dimensional materials are also examined. Finally, various methods for improving ohmic contacts’ characteristics, such as two-dimensional/metal van der Waals contacts and hybrid contacts, junction doping technology, phase and bandgap modification effects, buffer layers, are highlighted.

## 1. Introduction

Contacts, referring to a metal’s connection to a semiconductor or insulating film, are integral parts of all electronic devices. To exploit the electronic properties of electronic devices to a large degree, it is essential to produce ohmic contacts that allow for efficient and nondistorted signal transmission. Over decades of significant effort, solid theoretical accomplishments related to metal/semiconductor interfaces, and Schottky junction charge transport studies, as well as developments in the technological know-how for achieving good ohmic contacts, during the period from the 1930s to 1950s, contacts for microelectronic devices were not considered a major challenging issue. In the last couple of decades, contacts reappeared as a challenging issue with the aggressive downsizing of CMOS devices and the introduction of 2D materials, such as graphene, carbon nanotubes, and 2D transition metal dichalcogenides.

In the era of metal–oxide–semiconductor (MOS) or complementary MOS (CMOS) technology, many efforts have been devoted to the technology and the structural advancement of devices for the continual downsizing of the gate length and increasing of the integration density [1,2,3,4,5]. As the feature size of CMOS technology approaches a few nanometers, which is considered to be the physical and technological limits, shortening of the gate length has faced ever-tougher challenges and the pace of downsizing has slowed down in recent years (see Figure 1). The scaling rule and technology node assignment have shifted from physical gate length to equivalent gate length [6,7,8]. For instance, for the 130 nm technology node, the physical gate length is 65 nm, and the half-pitch width is 150 nm, implying that the widths of the spacer and the source–drain regions are approximately 85 nm. For technology beyond the 45 nm technology node, the gate length is 38 nm, and the half-pitch width is 68 nm [5], indicating a more aggressive scaling of the contact and spacer regions in later technology nodes. The source–drain contact regions have been narrowed to reduce the half-pitch size of the MOS transistor even when the gate length is not scaled according to Moore’s Law. With the FinFET technology introduced for the 28 nm technology node, the technology node assignment was based on the density level rather than the physical gate length. The 3D structure of FinFET and the larger effective gate width with a small footprint enables a higher chip density [8,9,10]. More nonclassical or non-Denard strategies [4], such as GAA/nanosheets, CFET, contact size reduction, cell size reduction, back power rail, buried power rail, back interconnection, nano TSV, and stacking or heterogeneous 3D packaging [11] (see Figure 1), will eventually be adopted to extend Moore’s Law to the subnanometer range.

Figure 2 shows the trend in contact scaling toward 1 nm technology. If we assume the spacer width is 8 nm, then one can estimate that the contact width for 5 nm technology is approximately 22 nm, which is the most substantial area of the whole transistor. According to the technological roadmap, as expected from the ITRS, the spacer width will be 6 to 5 nm in the coming technology nodes. An aggressive size reduction of the contact area is expected. The source and drain contact areas of FinFET structures have decreased from more than 1300 nm^2^ to approximately 500 cm^2^, respectively, in the downsizing of the 20 nm technology node to the 5 nm technology node [12]. However, to maintain the performance gains achieved with FinFET technology below the 40 nm CGP range, the contact resistivity should be smaller than 10^−9^ Ω-cm^2^. This is a challenging issue. A metal/semiconductor contact based on thermionic emission does not seem to be able to provide such a low resistivity value. In addition, it was predicted that for 30 nm CGP technology, the gate stack’s metal film thickness also needs to be scaled down to 6 nm [13], and the source/drain contacts for the nanosheet channel should be limited to the several decananometer range. The surface roughness of the thin films and their contact with silicon will have some adverse effects. We provide an in-depth discussion of these issues in Section 3.

On the other hand, two-dimensional materials are considered promising silicon replacements when silicon-based CMOS devices reach their physical limits [14,15,16,17,18,19,20]. A recent focus related to 2D materials are transition metal dichalcogenides (TMDs), such as molybdenum disulfide (MoS_2_) [21,22,23], ditelluride (MoTe_2_) [24,25], tungsten diselenide (WSe_2_) [26,27], and phosphorene. These emerging 2D materials have an acceptable gap value, which is unavailable in the first 2D material discovered: graphene. Like graphene, each monolayer of these materials is connected via the van der Waals force. These materials exhibit a number of attractive properties, such as high electron mobility, a low optical absorption coefficient, and high electrical and thermal conductivities. These advanced features could enhance the device and circuit performance in various aspects. However, there are still unresolved challenges related to mass production, complex device design, and circuit integration [20]. One of the challenges is achieving good ohmic contacts between 2D materials and metals. Unlike silicon, 2D materials have no dangling bonds and can form ideal contacts with metals. However, because only a few monolayers are thick and the dopant concentration is far less than that of conventional semiconductors, the contact resistances of metal/2D materials are much larger than those of conventional semiconductors. In addition, the new materials have different bandgap and band offsets, which require nonconventional metals to achieve the required energy level alignment for ohmic contact. In Section 4, we explore the physics of metal/2D material contacts, recent progress, and process options for tackling ohmic contact issues.

This review aims to provide a comprehensive overview of metal/semiconductor Schottky contact theory and its applications to 2D materials. Section 2 introduces the basic concepts, theoretical background, and assumptions used to develop the current–voltage model for Schottky contacts. We address some common misconceptions or inappropriate applications of the Schottky equation to the metal–insulator–semiconductor (MIS) structures. Section 3 examines the criteria and methods for achieving ohmic contacts and the scaling effects for contacts down to the decananometer range. Section 4 summarizes the recent advances in developing ohmic contacts for 2D materials, focusing on comparing metal van der Waals and hybridization contacts. We also review the techniques for junction doping, bandgap engineering, and phase engineering of 2D materials for contact optimization. Furthermore, we discuss the role of buffer layers in mitigating the Fermi-level pinning effects and enhancing the contact characteristics. Section 5 explores the characteristics of 2D material/Si contacts. Two-dimensional materials on silicon have been recognized as a feasible prior technology for the widespread application of 2D materials, which offers a promising way to leverage the advantages of both materials.

## 2. Schottky Junction

Long before the invention of the transistor, rectifying contacts were discovered and became a research hot spot [28]. It is known that the contact between a metal and a semiconductor has different resistance values depending on the polarity. This property led to the development of crystal rectifiers, such as copper–cuprous oxide rectifiers, which were widely used in early AM radio receivers for signal detection. Various theories have been proposed to model the rectifying characteristics. In particular, works by Nordheim [29], Frenkel [30], Fowler [31], Schottky [32], and Mott [33] have established a solid foundation for modern semiconductor physics, especially for current conduction across metal–semiconductor and metal–tunnel-oxide–semiconductor contacts. Schottky and Mott state that the first principle of band alignment and, thus, the barrier height is when two different materials are brought into contact. The rectifying contact resulting from a metal/semiconductor connection was later named a Schottky junction for the significant contribution of this widely accepted model.

### 2.1. Schottky Equation

The key component of the Schottky equation is the thermal emission law, as proposed by Richardson [34], which was based on experimental results related to the current density generated from a heated metal wire. The temperature and current dependencies are as follows:(1)$$J=A^{*}T^{2}\exp\left(\frac{W}{kT}\right)$$
where *W* is the workfunction of the metal, *kT* is the thermal voltage, and *A** is the effective Richardson constant. Schottky later modified the expression to include the effect of an electric field, and its final form is now usually expressed in terms of an applied voltage, *V*, as follows [32,33]:(2)$$J_{Schottky}=A^{*}T^{2}\exp\left(-\frac{\Phi_{\mathrm{SB}}}{\mathrm{kT}}\right)\exp\left(\frac{qV}{kT}\right)$$
where *q* is the electron charge, and *Φ_SB_* is the Schottky barrier.

In the original proposal by Schottky and Mott [32,33], the Schottky barrier is as follows”
*Φ_SB_* = *Φ_M_* − *χ*_S_(3)

Equation (3) provides the energy difference between the metal workfunction, *Φ_M_*, and the semiconductor affinity, *χ*_S_. The relationship is known as the Schottky–Mott rule, which describes the alignment of the energy bands when the two different materials have isolated state energies. For the case depicted in Figure 3a, an n-type semiconductor with an electron affinity smaller than the metal workfunction, equilibrium is achieved by transferring the electron on the semiconductor’s surface to the metal, as the metal workfunction is larger than the electron affinity of the semiconductor. The amount of band bending on the semiconductor’s surface is the same as the energy difference. However, the Schottky–Mott rule is not always accurate in predicting the barrier height. The Fermi level of the system tends to be pinned to a fixed position within the semiconductor bandgap, regardless of the metal used. This is because the charge distribution at the metal/semiconductor interface is not simply the sum of the charges on the isolated surfaces. There is also physical bonding or orbital overlap between the metal and semiconductor atoms, which changes the surface energy levels from their original values. This phenomenon is called Fermi-level pinning [35,36,37]. Equation (4) approximates the Schottky barrier height with the FLP effect.
*Φ_SB_* = (*S* × *Φ_M_* − *χ*_S_) + (1 − *S*)*Φ*_MIGS_(4)
where *Φ*_MIGS_ is the metal-induced gap states (MIGS). The intrinsic surface states and interface dipole of the semiconductor should lead to the same consequence. The parameter S is the pinning factor, which reflects the strength of the pinning and can be determined by the slope of the barrier height versus metal workfunction plot, expressed as follows:*S* = d*Φ_SB_*/d*Φ_M_*(5)

The barrier height does not depend on the metal workfunction when S = 0. This is the Schottky–Mott limit when S = 1. However, S is usually much smaller than one for most common semiconductors. Figure 4 shows how the barrier height changes with different metal workfunctions from various sources [36,37]. The slope deviates significantly from the ideal Schottky–Mott rule.

The effective Richardson constant was later treated as a universal constant or the fundamental Richardson constant, *A*_0_, which can be calculated by finding the ensemble of electrons leaving the metal surface at a kinetic energy exceeding the Fermi level of the metal, where the energy distribution of the electrons in the metal follows the Fermi–Dirac distribution. The fundamental Richardson constant is constituted by fundamental parameters, as follows:(6)$$A_{0}=\frac{4\pi qm^{*}k^{2}}{h^{3}}$$
where *m** is the effective mass of an electron, and *h* is Planck’s constant.

The fundamental Richardson constant, *A*_0_, equals 120 A/cm^2^/K^2^ when the effective mass is equal to the fundamental value of the electron mass. However, the effective Richardson constant, *A**, extracted from the experimental results using (2), always differs from the fundamental value, *A*_0_. In the Schottky diode, the thermionic emission from the semiconductor, the Richardson constant, is further modified to account for the effective mass of different bands and band structures of various materials [38]. However, experimental results over the last 120 years have not shown that *A** is a universal constant, even when taking the effective mass into consideration for the case of thermionic emissions from semiconductors. Instead, it is often treated as an empirical parameter or a fiction of *A*_0_ in many cases. Yet it is sound to consider the constant of a material-dependent parameter because of the different carrier velocities or lifetimes resulting from different or same materials prepared under varying conditions. In addition, the emitted electron may be reflected back to the electrode if we consider the carrier transport as wave propagation. Yet the aggressive scaling of the junction size in terms of the cross-section and thickness and the introduction of 2D material could offer an excellent opportunity to further disclose the mystery of the effective Richardson constant.

The second issue in applying the Schottky equation is that there is a vast list of publications, including numerous reports on 2D materials based on Schottky diodes or MIS Schottky diodes, that characterize the Schottky current with the “ideality factor”, *n*. Taking the ideality factor into consideration, the Schottky equation becomes [39], as follows:(7)$$J=A^{*}T^{2}\exp\left(-\frac{\Phi_{SB}}{kT}\right)\exp\left(\frac{qV}{nkT}\right)$$

The ideality factor was first defined in a pn junction diode, which has a solid physical ground, whereby if the forward current is contributed by the diffusion current only, *n* = 1, and when carrier generation and recombination (GR) take place, *n* = 2 [39]; *n* can increase up to 4 if the generation–recombination involves multiple energy levels of defects [40,41]. Thus, *n* = 1 represents an ideal diode, and *n* > 1 indicates the contribution of a GR current. The ideality factor indicates the quality of the pn junction diode. In a Schottky diode, although the introduction of *n* can help to improve the fitting of the voltage dependence in some cases, it is not an indicator of junction quality. It does not have a sound physical or technical meaning in many cases. The *n* factor in the Schottky diode equation should be a reflection of the effectiveness of the barrier surface potential varying with the applied voltage. We further elaborate on this issue in Section 2.2.

### 2.2. Current Conduction in an MIS Diode

The Schottky equation can be used to model the current conduction of an MIS diode. However, this equation is based on some assumptions that are often overlooked. It is also suggested that, in some early metal/semiconductor contacts, there may exist some native thin oxide on the semiconductor’s surface. Fowler suggested that there was a layer of “bad semiconductor” between a metal and a good semiconductor [31]. Although these proposals were later refuted, it did point to the importance of the interface’s quality in the contact behavior. A similar situation arose in the recent study of 2D material contacts and 2D material/Si contacts, in which a thin tunneling layer may be present between the metal and the 2D material. In this case, the Schottky equation can still be used to approximate the tunneling current in the forward region but with a modified effective Richardson constant that accounts for the tunneling barrier [42]. The modified Schottky equation is as follows:(8)$$J_{MIS}=A_{0}T^{2}\exp(-\sqrt{\chi }\delta )\exp\left(-\frac{\Phi_{SB}}{kT}\right)\exp\left(\frac{qV}{nkT}\right)$$
where is the χ mean barrier height value between the insulator and the semiconductor and *δ* is the tunneling oxide thickness. We can define a new effective Richardson constant as follows:(9)$$A^{*}=A_{0}\exp(-\sqrt{\chi }\delta )$$

Note that the current conduction is, in fact, dominated by the direct tunneling of the carriers over the thin insulating layer. The approximation of the Schottky equation involves modifying the Richardson coefficient with a transmission coefficient, which depends on the barrier height between the silicon and the insulator and the insulator thickness. These two factors are the key parameters that govern the conduction current. In addition, some effects, such as the minority carrier injection, image force, and quantum mechanism reflection, are also neglected. This approximation is only valid for a forward current with *V* > 3*kT*/*q*, because the metal/insulator barrier differs from the silicon/insulator barrier. Some authors have ignored these assumptions and treated the MIS diode as a regular Schottky diode.

Instead of treating the *n* factor as an “ideality factor” in (8), it was proposed that the nonunity *n* factor could be calculated from the surface potential of the barrier [42]. Defining *n* as a change in the surface potential, Δ*V_s_*, with respect to the applied voltage (i.e., $n=-V/\Delta V_{s}$), and further taking the interface states into consideration, the electrostatic calculation leads to the following:(10)$$n=1+\frac{\left(\frac{\delta }{\varepsilon_{i}}\right)\left(\frac{\varepsilon_{s}}{W}+qD_{s}\right)}{1+qD_{m}\delta /\varepsilon_{i}}$$
where *ε_i_* and *ε_s_* are the permittivities of the insulator and semiconductor, respectively; *W* is the depletion layer in the semiconductor surface; and *D*_s_ and D_m_ are the surface state densities in the equilibrium with the semiconductor and metal, respectively. If the surface state is negligible, (10) reduces to the following:(11)$$n=1+\frac{\delta /\varepsilon_{i}}{W/\varepsilon_{s}}$$

Thus, large n values, in some cases, are not mysterious; it is only because the electrostatic thickness of the insulating layer, $\delta /\varepsilon_{i}$, is much larger than the electrostatic thickness of the depletion layer width, $W/\varepsilon_{s}.$ If we neglect the surface states on the metal side, which is the case for most MIS diodes, Equation (10) reduces to the following:(12)$$n=1+\frac{\delta }{\varepsilon_{i}}\left(\frac{\varepsilon_{s}}{W}+qD_{s}\right)$$

In this expression, n is a parameter that describes how effectively the barrier surface potential varies with the applied voltage. It has nothing to do with the generation and recombination of carriers or the validity of the Schottky current model. Therefore, it is not an indicator of the ideality of the junction.

In fact, the validity of (8) for the MIS diode is limited. The conduction current should be more precisely modeled by the direct tunneling current (see Equation (13)) if the insulator layer and barrier height are in the direct tunneling range [43,44].
(13)$$J_{DT}=J_{0}\left(1-\frac{V}{2\Phi_{B}}\right)exp\left\{-\frac{4}{3}\frac{\sqrt{2m_{i}q}\delta }{\hbar }\frac{\Phi_{B}}{V}\left[1-{\left(1-\frac{V}{\Phi_{B}}\right)}^{3/2}\right]\right\}$$
where *m_i_* is the electron mass in the oxide; ℏ is the reduced Planck’s constant; and *Φ_B_* is the barrier height (in electron volts) between the emitting electrode and the oxide.

For the case of thicker insulators or with a smaller voltage applied, tunneling over the triangular edge (see Figure 3c) is also possible. In this situation, the current conduction is better described with the Fowler–Nordheim (FN) equation, as shown below [45]:(14)$$J_{FN}=AE^{2}\exp\left(-B/E\right)$$
(15)$$A=\frac{q^{3}}{8\pi h\Phi_{B}}\left(\frac{m_{e}}{m_{i}}\right)$$
(16)$$B=\frac{4}{3}\frac{{\left(2m_{i}\right)}^{1/2}}{q\hbar }{\Phi_{B}}^{3/2}$$
where *m_e_* is the electron mass in the free space.

The FN formula has been widely used to explain the conduction behavior of various thin dielectric films [45]. However, in contact studies including the so-called MIS Schottky diode, the current–voltage characteristics were often fitted with the modified Schottky Equation (8) or even the simple Schottky equation in (7) instead of the FN relationship. It has to be pointed out that the barrier height extracted from these fitting should be inaccurate, and the ideality factor does not carry any technical implications. The effective Richardson coefficient is also inappropriate in these cases. The current–voltage characteristics are better described with the FN equation.

## 3. Ohmic Contact in the Nanoscale

Ohmic contact refers to a metal–semiconductor contact with low and constant resistance regardless of the applied voltage polarity. A low contact resistance and nonrectifying contact enable signals to transmit into and out of a semiconductor device, such as transistors, LEDs, and solar cells, with minimum distortion. Figure 5a,b illustrate the idea for achieving an n-type ohmic contact. A small contact barrier can be obtained by choosing a small metal workfunction. Under forward bias, the electron can be emitted through the barrier via thermionic emission or Schottky emission. With reverse bias (see Figure 5b), the barrier height is increased, but a large reverse current is still possible if the barrier width is narrow enough so that direct tunneling is possible. The ohmic contact has not been considered a server-challenging issue since the start of silicon technology. The key factor governed by the contact resistance can be readily solved with the well-developed metallization and junction doping technique. The narrow barrier is achieved by heavily doping the contact region of the semiconductor. In addition, the junction dimensions, including the cross-sectional area, junction depth, and metal thickness, are quite large compared to the devices themselves. Figure 5c illustrates the ideal ohmic contact’s characteristics, practical ohmic contact, and the scaling effect of the contact. As mentioned, to reduce the signal loss, the contact resistance should be as small as possible. The current–voltage (I–V) relationship in the prescribed voltage range should be linear so that it does not cause signal distribution and harmonics. However, as illustrated in Figure 5a,b, barriers exist in the metal/semiconductor contact; the I–V characteristics are either governed by thermionic emission or tunneling, which are nonlinear. In addition, because the forward and reverse conductions are controlled by different mechanisms, the I–V characteristics are asymmetric. The degrees of nonlinearity and asymmetry may not cause significant issues when the signal levels are high. However, they could significantly downgrade the device’s performance when the signal level is low. The contact’s characteristics will further seriously deteriorate if the contact size is reduced, the signal level is lowered, and, in some cases, new materials and fabrication processes are introduced in nanoscale CMOS technology.

Figure 6a shows the key parameters and process issues for producing a good ohmic contact. Generally, ohmic contact has not been a major challenge in CMOS technology over the past six decades. The contact size is still quite large compared to the conductive current and applied voltage. Moreover, a suitable metal workfunction is available, and the semiconductor can be doped to the degeneracy level. However, in the nano CMOS technology, the situation is changed. To reduce the chip size, every dimension has been scaled down. In the “2 nm” technology, even the power rails of the integrated circuit are scaled and moved to the back side of the IC. Reducing the contact’s cross-sectional area is undoubtedly also an attractive option for fitting more transistors in the chip. Some new technology options, such as the use of high-k gate dielectric materials, will limit the choice of materials for metal gates [6]. Introducing a nanosheet device structure will result in an ultra-shallow junction, and heavy junction doping may not be feasible. The thickness of the metal layer also needs to be reduced. Consequently, the contact interface and the surface roughness of the metal film can become significant factors affecting the contact quality. The process consequences of contact scaling for nano CMOS technology are illustrated in Figure 6b.

Figure 7 shows how the physics of a Schottky junction and an ohmic contact can be affected by scaling down the junction size. The main effects are as follows:
(1)Barrier lowering: The barrier height at the interface can be reduced by various factors, such as image force effects, metal workfunction variation, and surface roughness. These factors may be negligible in a large junction, but they can have significant impacts in a scaled junction. They can increase the forward current by lowering the potential barrier.(2)Barrier widening: The use of heavily doped contacts may not be possible in a scaled junction. This can result in a wider tunneling barrier, which reduces the conduction current under reverse bias.(3)Richardson constant reduction: Thinner metal films are used in a scaled junction, which can lead to a smaller value of the Richardson constant. This can decrease the forward current by reducing the thermionic emission.(4)Metal workfunction lowering: experimental results suggest that thinner metal films have lower workfunctions [46], which can also reduce the barrier height at the interface.(5)Interface states: The presence of interface states can cause Fermi-level pinning, which affects the barrier height and the band bending. This effect is more pronounced in thinner films and in unpassivated surfaces.(6)Interface layer: To mitigate the Fermi-level pinning effect or to enable different circuit design options, such as using 2D materials as interlayer conductors (see Section 4.5), an interface layer may be used for passivation. This can affect the reverse currents by changing the tunneling characteristics. The thickness, band offset, and dielectric constant of the interface layer are important parameters for this effect.

### 3.1. Effects of Junction Doping

The unavailability of a heavily doped contact results in a wider barrier. As shown in Figure 7a, this reduces the tunneling efficiency. Following the method developed by Yu [47], we can calculate the contact resistance depending on the dopant concentration. Figure 8 shows the contact resistance as a function of the dopant concentration. For high dopant concentrations (>2.5 × 10^19^ cm^−3^), the contact resistance decreases to the range of 10^−5^ to 10^−6^ Ω-cm^2^. The current conduction is mainly due to field emission. For low-level doping (<10^18^ cm^−3^), the contact resistance increases by 5 to 6 orders of magnitude. The current conduction is mainly governed by thermionic emission. In the intermediate doping range, the current conduction is a combination of thermionic and field emissions. Hence, one should be aware that the Schottky equation is not always valid, especially when the doping concentration is high. The field emission needs to be taken into account. On the other hand, the contact resistance can be significantly reduced when field emission takes place. Thus, in a nanoscale contact for a 30 nm CPP [12], the ohmic contact regime should shift from the thermionic region to the field emission and tunneling regimes.

### 3.2. Effects of Interface Roughness

The Schottky current of a metal/semiconductor can exceed the Schottky–Mott limit. Fundamental calculations show that the metal surface flatness affects the electron cloud near it [48,49]. It was found that the ratio of electric field fluctuations to the average electric field, denoted by δ*E*/*E_S_*, can be estimated by the following [49]:(17)$$\frac{\delta E}{E_{S}}=\Delta^{2}\int_{0}^{k_{c}}\frac{{\left(l_{cor}k\right)}^{2}}{{\left(1+a\left(l_{cor}k\right)\right)}^{1+r}}dk$$
where *E_s_* denotes the electric field resulting from a smooth surface; δ*E* is the increased electric field due to the surface roughness; *k* is related to the surface wave vector; Δ is the normalized roughness; *l_cor_* is the normalized correlation length; *r* is the roughness exponent, which is a measure of the degree of surface irregularity; and *a* is a proportional constant. See Figure 9 for the definitions of the parameters.

Figure 9 shows how the roughness parameter, Δ, and the correlation length, *l_cor_*, are normalized by the film thickness, *t*_diel_. The normalized roughness Δ (=r_s_/tdiel) becomes more important for films with similar thicknesses. The correlation length, which measures the local field variation, is also inversely related to the film thickness, i.e., *l_cor_* = *L_cor_*/*t*_diel_. This means that thinner films have larger values of Δ and *l_cor_* and, therefore, larger electric field fluctuations [50]. However, the surface roughness is not a scalable parameter. When the film thickness reaches the atomic scale, the cluster size of a polycrystalline structure affects the surface roughness negatively. The increased local field lowers the effective barrier of the contact, which results in a significant rise in the Schottky emission, Fowler–Nordheim (FN) tunneling, and Poole–Frenkel (PF) emission [49,50].

One of the factors that affect the electrical properties of metal–insulator–metal (MIM) structures is the surface roughness of the interfaces. The lower interface (insulator on metal) tends to be rougher than the upper interface (metal on insulator) due to the crystalline nature and the grain size of the metal films, as well as the limitations of the deposition methods, such as evaporation or sputtering. This phenomenon has been known for decades and persists even in MIM structures with larger dimensions and thicker films [51,52,53,54,55,56]. Figure 10 shows an example of how the surface roughness influences the current conduction of MIM structures. The experiment was performed on TiN/Al_2_O_3_/TiN MIM capacitors with identical top and bottom electrodes. Ideally, the Schottky barriers should be equal, resulting in a symmetric current–voltage characteristic. However, Figure 10b reveals that the measured I–V characteristic is asymmetric. This is because the bottom and top barriers have different values: 3.01 eV and 3.65 eV, respectively (see Figure 10a).

The electrical properties of metal/semiconductor contacts strongly depend on the metal film thickness and, also, the method of deposition [46]. These effects may be partly related to the surface roughness. Figure 11 shows how the Schottky barrier height and Richardson constant vary with the metal thickness and deposition process. For an evaporated sample with a 100 Å thickness (see Figure 11a), the Cu electrode has a barrier height of approximately 0.58 eV and a Richardson constant close to the theoretical value of 112 A/cm^2^/K^2^. Both parameters increase sharply and reach their saturation values of about twice the theoretical value for the Richardson constant and 0.62 eV for the barrier height for films thicker than 200 Å. For the sputtered Cu film (see Figure 11b), the theoretical Richardson value is maintained up to 200 Å. It increases almost exponentially as the film becomes thicker. The Richardson constant rises to about 1700 A/cm^2^/K^2^, which is approximately 15 times the theoretical value, and the barrier height to 0.71 eV for 800 Å thick Cu films. Toyama attributed the difference to the high kinetic energies of Cu atoms during sputtering but did not provide further explanation. It is unclear how the kinetic energies of metal atoms can affect the Schottky characteristics. Unlike thermal evaporation, sputtering involves metal atoms with much higher kinetic energy. However, these results should be caused by other factors. Generally, the sputtering process produces a smaller grain size of the deposited metal clusters than the evaporated ones. Another important factor is that thin film sputtering is usually conducted in a low vacuum, resulting in a higher oxygen content in the film than evaporation. Evaporated films have larger grain sizes than sputtered films, which leads to a higher surface roughness. This could account for the lower barrier height observed in the evaporated films than the sputtered ones. The same mechanism can also explain the effect of the film thickness. A thicker film would have a smoother surface and, thus, a higher barrier height.

Furthermore, because of the high oxygen content in the sputtered films, the electrons’ film will be strongly bounded to the oxygen because of the greater electronegativity of the oxygen; as a result, it makes the thermionic emission of electrons more difficult. Theoretical calculations have shown that the workfunction of the Cu-O system is larger than that of pure copper [57]. The experimental results show that both the barrier height and the effective Richardson constant decreased. Factors such as the barrier height inhomogeneity and effective mass variation may have some effects [36,37], but they cannot, however, account for such a large difference in the effective Richardson constant. Because the effective Richardson constant has the same trend as the barrier height in terms of the thickness dependencies, one may infer that the effective Richardson constant is a function of the barrier height. Nevertheless, these results imply that the conduction current of an ultrathin metal/semiconductor contact at low voltage is lower than that of a thick metal contact. However, at high voltage, the situation is reversed, and the thin contact has a higher conduction current than the thick contact. Therefore, the nonlinearity of the I–V characteristics will be further deteriorated. The process dependence also becomes more significant as smaller size contacts and thinner metals are used. In the nano CMOS process, the metallization will gradually be replaced by the atomic layer deposition (ALD). The ALD process can produce films with high thickness uniformity and conformal coverage. However, ALD metal films are usually polycrystalline and have large grain sizes. This implies that the surface roughness will be high if the film thickness and contact cross-section are in the order of several hundred nanometers [58]. Figure 12 illustrates some examples of copper films deposited by the ALD process; in the worst case, the RMS roughness is 21 nm for a 4.7 nm thick film. This high roughness of the metal film would result in significant barrier lowering and lead to a large fluctuation in the contact resistance. Thus, preparing a metal film with low roughness could be the key issue for smaller-sized contacts.

In summary, contacts have become one of the major challenges in nano CMOS technology. The challenge comes from the requirement of achieving an even lower contact resistance with the aggressively reduced contact area and film thickness. This challenge is exacerbated by the device structure’s evolution, such as the transition from planar to FinFET and nanosheet architectures. Moreover, the fabrication process changes, such as the introduction of high-k/metal gate stack and the thin film deposition techniques, also pose difficulties for contact engineering in nano CMOS technology.

## 4. Contacts for 2D Materials

Two-dimensional materials, especially transition metal dichalcogenides (TMDs), such as molybdenum disulfide (MoS_2_) [21,22,23], ditelluride (MoTe_2_) [24,25], tungsten diselenide (WSe_2_) [26,27], and phosphorene, are considered to be possible replacements for silicon when silicon-based CMOS devices reach their physical limits [15,16,17,18,19,20]. These materials, which have high electron mobility, low optical absorption coefficient, and high electrical and thermal conductivities, could enhance the device and circuit performance in many aspects. However, there are still unresolved challenges related to mass production, complex device design, and circuit integration [15,16].

Most of the reported 2D-material-based devices are currently much bigger than CMOS technology, even though 2D materials are often linked to nanodevices and nanotechnology in the literature. Achieving a good ohmic contact at the submicrometer scale is still a difficult problem. Because the films are only a few monolayers thick and the dopant concentration is far less than that of conventional semiconductors, the contact resistances of metal/2D materials are much larger than that of conventional semiconductors (see Figure 13). Shen et al. reported an ultralow resistance ohmic contact based on semi-metallic bismuth on monolayer TMD materials. The Schottky barrier height was reduced to a zero voltage [59]. There is a large body of literature on 2D-material-based Schottky contact devices from over the last decade [60,61,62,63,64,65]. Many results have been obtained using the simple Schottky–Mott rule, neglecting the technological issues and other mechanisms related to the charge transport over the barrier, resulting in inaccurate and inappropriate results for the barrier height and pinning parameter. For example, the wide scatter of the pinning parameter versus metal workfunction plot, as shown in Figure 14, for the 2D material, provided by Wang et al. [62], is an indication of the improper treatment or interpretation of the experimental data from this simple junction. Although 2D materials have atomically flat surfaces and minimal dangling bonds or charge traps, which are advantageous for ohmic contact formation, 2D materials are difficult to dope heavily, and metal/2D material contacts are not as good as metal/Si contacts. The contact resistance is high even with a much larger surface area. These issues need to be addressed before 2D-material-based devices can be scaled down to the nanoscale. The high contact resistance and the nonlinear current–voltage characteristics of the contact could limit low-voltage and high-performance circuit applications.

Figure 15 shows the main features and challenges of metal/2D material contacts, which are summarized as follows:
(1)van der Waals gap: This is a tunneling barrier between the metal and the 2D material that allows for the tunneling of electrons. Some 2D materials may also form strong bonds with the metal or by overlapping their orbitals.(2)2D contact: This is a common method of connecting 2D material from the top, but it has a high resistance per area because the current flows parallel to the 2D plane, not perpendicular to it. A top contact is easier to achieve because it involves depositing metal on the surface of the 2D materials and patterning it with standard photolithography.(3)Hybridization and edge contact: This is a better alternative to the 2D contact, as it creates a physical bond and a direct current path along the surface of the 2D material. However, it is challenging to achieve, because it requires the precise alignment and deposition of metal on very thin edges. Some 2D materials may also form strong bonds with the metal or by overlapping their orbitals.(4)Doping: 2D materials cannot be doped in the conventional ways, and it is hard to dope heavily.(5)Metal intercalation: This is a process of incorporating metal atoms into the gaps of multilayer 2D materials. The dopants contribute to the current’s conduction and can improve the contact’s conductivity.(6)Surface defects: These are imperfections, such as sulfur (S) vacancy in MoS_2_, on the 2D material surface that can trap charges and pin the Fermi level, affecting the contact potential and resistance.(7)Layer-dependent bandgap: The bandgap and the contact potential of the 2D material vary with the number of layers. This can be exploited to tune the contact properties by changing the layer thickness in the contact region.(8)Phase modification: Some 2D materials can switch among different phases that have distinct electrical properties. For instance, the 2H phase of TMDs is semiconducting, while the 1T and 1T0 phases are metallic. By changing the contact region to a metallic phase, the contact conductivity can be enhanced significantly.(9)Buffer layer insertion: inserting a buffer layer between the metal and the 2D material can help reduce the effects of the van der Waals gap and the metal-induced gap states.(10)Metal workfunction selection: To achieve good ohmic contact with both n-type and p-type 2D materials, various unconventional metals, such as In, Mg, Ag, Pd, Sc, and Ti, have been explored. However, not much work has addressed the issues of stability, reliability, and potential process contamination. For digital circuit applications, one must consider whether these metals can produce the desired threshold voltages for n-type and p-type transistors.

In the remaining part of this section, we further elaborate on these issues. The recent progress on techniques to improve these 2D materials/metal contacts is highlighted. The discussion is categorized into five subsections: (1) metal van der Waals contacts and hybridization contacts; (2) junction doping; (3) bandgap modification; (4) phase modification; (5) Fermi-level pinning and buffer layers.

### 4.1. Metal van der Waals Contacts and Hybridized Contacts

One of the challenges in fabricating devices based on 2D materials is to ensure a good contact between the 2D lattice and the 3D metal electrodes, without damaging the 2D structure or creating interface defects. To address this issue, some researchers have developed techniques to integrate 3D metal layers with 2D materials using van der Waals (vdWs) forces. These techniques preserve the integrity of the 2D lattice and avoid chemical bonding. For example, Liu et al. [63] devised a novel method to transfer atomically smooth metal layers onto 2D semiconductors. The procedure involved patterning and depositing the metal layer on an atomically flat Si substrate, then covering it with a PMMA film to enable its detachment. Afterward, a PDMS stamp was employed to lift the PMMA-coated metal layer and align it precisely on the target surface of the 2D material. On the basis of this method, the Schottky barrier height obtained was close to the Schottky–Mott model and had a large pinning factor of 0.96 (see Figure 16). In contrast, when the metal contact was made using direct evaporation, the barrier height was much smaller, and the pinning factor was reduced to 0.09. However, the metal transfer method is not a mass production technique. For an actual integrated circuit, the surface profile will be highly uneven, and the coverage and air gaps on the step edges will be an unresolvable issue. Wang and coworkers also presented a simple technique to produce van der Waals contacts on single-layer MoS_2_ using Au-capped In as the contact material, and the contact resistance is around 3.3 ± 0.3 kΩ·mm [64].

From a fabrication process and device operation point of view, there should be a better approach to achieving covalent bonding between 2D materials and metals. An intimate contact would help to eliminate the interfacial tunneling barrier and enhance the carrier injection efficiency. Side or edge contacts on 2D materials are one way to obtain these kinds of metal/2D material contacts. The edge contacts can be fabricated by using either top-down or bottom-up approaches. In a top-down scheme, an insulating layer, such as Al_2_O_3_ or h-BN, is first deposited on the 2D channel. It is then selectively etched to expose the 2D edges. It was reported that edge-contacted graphene FETs can have a contact resistance as low as 100 Ω·mm while maintaining a high charge carrier mobility [66]. Yang et al. also demonstrated 1D edge contacts on few-layer MoS_2_ and found a high pinning factor of ~0.975 for MoS_2_-based FETs with different edge contact metals [67]. The bottom-up approach involves growing 2D materials from 3D metal seeds or another 2D metal material. This method can produce high-quality and uniform edge contacts, which improve the conductivity and mobilities of TMD-based devices [68,69].

Some metal/2D material contacts are not thermally stable. The contact characteristics are affected by the interface interaction. McDonnell et al. observed a thin layer of TiO_2_ when Ti was deposited on MoS_2_ with a vacuum pressure of 10^−6^ mbar. This layer was found to have a negligible conduction band offset with MoS_2_ and resulted in a low contact resistance [70]. However, when Ti was deposited in an ultrahigh vacuum of 10^−9^ mbar, the Ti atoms reacted with MoS_2_, and a Ti_x_S_y_ phase formed, which resulted in a much larger contact resistance. The formation of TiO_2_ should be due to the residual oxygen or water molecules in the vacuum system or metal. The thin TiO_2_ layer inhibits the direct chemical reaction of Ti and S. At an ultrahigh vacuum, the oxygen and water residuals are significantly reduced, and the direct reaction of Ti and S is possible. Ti_x_S_y_ has poor electrical conductivity. However, Au/MoS_2_ contacts were also found to be dependent on the deposition pressure. English et al. found that Au contacts deposited at 10^−9^ torr had the lowest contact resistance of 740 Ω⋅mm, which is three times smaller than those deposited at 10^−6^ torr [71]. Yet this difference can also be attributed to a reduced impurity absorption at the interface under high vacuum. The different surface roughnesses of the as-deposited Au films prepared by other deposition pressures should be the major cause of the different contact resistance values.

To realize CMOS devices with 2D-material-based transistors, achieving both n-type and p-type devices on a single-type 2D material and with the same metal contacts is essential. It was discovered that the polarity of multilayer MoTe_2_ encapsulated in h-BN can be varied by applying thermal annealing at different temperatures [72]. Liu and coworkers found that the MoTe_2_ device changed from a p-type to an n-type conduction after annealing in N_2_ at 150 °C. In particularly, the electron Schottky barrier decreased to 340 meV, and the hole barrier increased to 560 meV [72]. They attributed these changes to the annealing-induced interfacial bond hybridization of the Au/MoTe_2_ contacts. Hence, an interface reaction or interface hybridization could be a way to reduce the contact resistance of 2D/metal contacts. It is worth a comprehensive study.

### 4.2. Junction Doping

High substrate doping has been the key strategy in conventional semiconductor technology to achieve ohmic contacts in CMOS technology. Conventional semiconductors can use different doping techniques, such as ion implantation and plasma immersion ion implantation [1], to increase the doping concentration to a degeneracy level. However, these methods are not compatible with 2D materials that have only a few atomic layers. A more effective doping method for 2D materials is spontaneous charge transfer doping (SCTD) [73,74]. For p-type doping, the energy difference between the top of the valance band of the 2D material and the lowest unoccupied molecular orbital (LUMO) of the surface doping will cause electron transfer, which leads to hole accumulation and the lifting upward of the valence band in the semiconductor surface region. Similarly, suppose the bottom of the conduction band of the 2D material is below the highest occupied molecular orbital (HOMO) of the surface dopant; electrons will transfer from the HOMO of the surface dopant to the conduction band of the semiconductor and result in the electron doping. A wide range of dopants for various 2D materials are available [73]. For instance, by immersing WS_2_ or MoS_2_ in pure 1,2-dichloroethane (DCE) at room temperature, n-type doping with chlorine atoms was obtained [74]. The doping was due to filling sulfur vacancies with Cl in these 2D materials. With Ni metal contacts, the contact resistance values of the doped devices decrease significantly (see Figure 17) to 0.7 kΩ⋅mm and 0.5 kΩ⋅mm, respectively, for WS_2_ and MoS_2_. This improvement is attributed to the thinning of the Schottky barrier because of the high carrier concentration from the semiconductor’s side.

Plasma doping is another method to achieve contact doping for 2D devices [75,76]. Under hydrogen plasma irradiation, Tosun et al. demonstrated that Se vacancies are produced in WSe_2_ crystals [75]. As shown in Figure 18, after 12 s hydrogen plasma treatment, the E_F_ – E_V_ increased from 0.73 eV to approximately 1.19 eV, which represents a doping level of over 4 × 10^17^ cm^−3^ at room temperature. The near-degeneracy doping level leads to a better contact performance. For a thick device with ten monolayers of WSe_2_, the contact resistance is as low as 4.4 kΩ·μm [75]. Similarly, Kang et al. obtained ohmic contacts for p-type WSe_2_ FETs with oxygen plasma treatment [76]. The O_2_ plasma irradiation formed a WO_3_ intermediate layer with a larger workfunction, which provided a good contact for p-type WSe_2_ FETs. It was found that a focused light beam can also cause local oxidation of a MoTe_2_ film [77]. The oxidized film had a higher hole concentration, improving the p-type semiconductor/metal contact region. The optical technique has the advantage of precise control, but the increased hole level is still large enough for good ohmic contact. Other nonconventional doping techniques, such as ionic liquids and polymer electrolytes [78,79,80], do not seem suitable for mass production.

### 4.3. Bandgap Modification and Band Alignment

Band alignment can be achieved by selecting a metal with a suitable workfunction. Previous studies have shown that the contact potential depends on the type of surface termination [81]. For top contacts with a monolayer of MoS_2_, the Fermi-level pinning is close to the conduction band edge, and n-type Schottky barriers are formed (See Figure 19a). In this case, Al, In, and Mg may be good candidates for ohmic contacts. The edge contacts, either armchair or zigzag termination of Mo and S atoms, result in a Fermi-level pinning near the valence band, and p-type Schottky barriers are formed (see Figure 19b–d).

Two-dimensional material has a distinct nature in that the bandgap can be readily tuned by stacking a different number of 2D monolayers. The bandgaps of 2D materials vary with the number of layers and their stacking configurations, which influences the quantum confinement effects. On the basis of the first-principles calculations, Wickramaratne et al. found that the band gap of a single monolayer of hexagonal boron nitride (h-BN) is a direct bandgap semiconductor; the gap becomes indirect for multiple layers [82]. The positions of the band edges, with respect to the vacuum level, shift by 0.5 eV for the direct-to-indirect transition (see Figure 20). Thickness-dependent bandgap characteristics were also found in other 2D materials. A comprehensive study was conducted by Li and coworkers in an attempt to optimize the MoS_2_ field-effect transistor’s fabrication [83]. Figure 21a shows the bias dependency of area contact resistivity (ρ_c_) extracted from the MoS_2_ field-effect transistor with a different numbers of MoS_2_ monolayers [83]. A turn-around behavior on the number of MoS_2_ monolayers was observed (see Figure 21b). This observation was explained in terms of the different junction depths for a thicker MoS_2_, which affects the carrier distribution and injection path (see Figure 21c). However, it is not sufficient to explain the contact resistivity change unless the barrier height increase is considered. As shown in Figure 21e, the slope of the resistivity changes with the film thickness, and modifying the barrier height will result in a better fitting. The change in the barrier height results in different carrier injection mechanisms: thermal emission and thermal field emission resulting from the different values of the band offsets. The thickness-dependent bandgap (*E*_g_) and the barrier height (*ϕ*_B_) of Au/MoS_2_ contacts are shown in Figure 21f, and the energy-level alignment at Au/MoS_2_ interfaces for different MoS_2_ thicknesses is shown in Figure 21g. The barrier height dropped from 0.65 eV to 0.33 eV, leading to a higher contact resistance, as the MoS_2_ thickness grew from one monolayer to five monolayers. However, for MoS_2_ thicker than five monolayers, the trend changed. The material began to act more like a bulk material than a 2D material, and the inactive layers increased the contact resistance. These results suggest a simple way to reduce the Schottky barrier by choosing 2D materials with a suitable thickness. This unique feature leads to an important processing option, which should have several potential applications for future device technology. For example, it enables bandgap and barrier engineering for devices such as Schottky diodes, photodetectors, or solar cells. It can also allow for the customization of the channel materials for field-effect transistors. Moreover, as this review discusses, it can provide a technological solution for contact engineering. This unique feature is not possible in traditional CMOS technology.

### 4.4. Phase Modification

Two-dimensional materials can exist in different crystal structures or phases. Different phases have different electronic and optical properties and, thus, different metal contact behaviors as well [84,85]. For example, 2H-TMDs are semiconducting materials with a trigonal prismatic structure, while 1T and 1T0 phases are metallic materials with an octahedral structure. Using n-butyllithium to induce a phase transition from 2H to 1T’ or 1T0 in a MoS_2_ nanosheet, Kappera et al. achieved a significant reduction in contact resistance to 0.24 kΩ⋅μm [85]. Sun et al. also reported similar results with a 1T’ phase contact in MoS_2_ [86]. In contrast, molybdenum ditelluride (MoTe_2_) has a very small energy difference (~35 meV) between the 2H and 1T0 phases in a single layer, which enables a low barrier metal/MoTe_2_ contact with the 1T0 phase [87]. Qi et al. demonstrated an ohmic contact with a very low Au/MoTe_2_ barrier height of 10 meV with a thermal anneal-induced phase transition [88]. Using a laser-induced phase transition, the metallic monoclinic 1T’ phase of MoTe_2_ was obtained from the 2H phase MoTe_2_ by Cho et al. [89]. It was reported that the barrier can be reduced from 0.2 eV to 0.01 eV by changing the 2H contact to a 1T’ contact [89]. The significantly reduced Schottky barrier leads to high conductivity and an increase in the carrier mobility of the MoTe_2_ transistor by 50 times [89].

### 4.5. Fermi-Level Pinning and Buffer Layer

The interface states at two-dimensional material/metal interfaces can affect the performance of two-dimensional devices. These states include metal-induced gap states (MIGS) and defect states, which can induce Fermi-level pinning (FLP) at the interface and hinder the carrier injection efficiency. Fermi-level pinning causes adverse effects in 2D material/metal contacts. It was proposed to be one of the main reasons for the experimental observations of large contact resistance. The origins of the FLP effect at the 2D material/metal interface should be due to several causes. As mentioned, the barrier height of a metal/semiconductor contact can be tailored by the proper choice of metal with the desired workfunctions according to the Schottky–Mott rule. However, the principle is not always valid because of the Fermi-level pinning effect. Similar to Fermi-level pinning in a high-defect Si/metal oxide interface, intrinsic defects or impurities, such as sulfur vacancies or surface metal-like impurities on TMD materials, can act as electron donors [90]. In addition, various physical or chemical processing steps during a device’s fabrication can also produce various types of damage to the crystalline structure of 2D materials or trigger interface chemical reactions, which are additional regimes for the Fermi-level pinning of a contact.

Fermi-level pinning is often considered to arise from metal-induced gap states (MIGS) as a result of the metal electronic wave function overlapping with the 2D material at the charge neutrality level of the semiconductor. Guo et al. [91] computed the Schottky barrier of various metals on transition metal dichalcogenides (TMDs) without intrinsic defects and found a strong pinning effect with a pinning factor of S = 0.3. They attributed this result to the strong chemical bond between metals and chalcogens [91]. This type of FLP effect was verified by scanning tunneling microscopy (STM) experiments and theoretical models [92]. Kerelsky et al. showed the existence of MIGS in MoS_2_ using scanning tunneling spectroscopy (STS) to measure the local density of states (LDOS) along the 2D/metal interface. The LDOS gap decreases as the contact edge becomes closer [92].

A buffer layer can reduce the effects of metal-induced gap states (MIGS) by creating a matching layer between the 2D material and the metal. This idea was inspired by previous work on high-performance metal–insulator–semiconductor (MIS) diodes [33,42]. The buffer layer prevents the metal wave function from penetrating the semiconductor material, which lowers the MIGS density. Moreover, if the buffer layer is an insulator, it can balance the charge at the interface and shift the Fermi-level closer to the charge neutrality level, which further decreases the effective Schottky barrier height. However, as given in Figure 7, the buffer layer should be thin enough to allow for efficient tunneling and have good interface quality with both the 2D material and the metal. The buffer layer can be metallic, semiconducting, or insulating.

One of the most often used buffer materials for 2D material/metal contacts is graphene. Graphene has a semi-metallic nature and a tunable workfunction. By inserting prepatterned single-layer graphene between MoS_2_ and metal, Chee and co-workers created a sandwich contact structure for MoS_2_ FETs [93]. The graphene layer enhanced the contact between MoS_2_ and Ag significantly. The Schottky barrier height was reduced to 190 meV due to the charge transfer from Ag to graphene that matched the Fermi level of graphene with a conduction band edge of MoS_2_. This increased the electron mobility by almost three times. Graphene is not the only metallic 2D material that can act as an interfacial buffer layer to achieve ohmic contacts for 2D-material-based devices. Other examples include NbS_2_, WTe_2_, and heavily p-doped Wse_2_ or MoS_2_ [94,95,96]. Metal oxides, such as TiO_2_, Ta_2_O_5_, MgO, and ZnO, are also suitable for metal/insulator/2D semiconductor structures [97,98,99]. A buffer layer thickness dependence study was conducted for the MoS_2_/Ta_2_O_5_/Ti structure. By inserting a Ta_2_O_5_ layer with different thicknesses up to 5 nm for the MoS_2_/metal contact, Lee et al. [99] found that the contact resistance first decreases as the Ta_2_O_5_ buffer layer increases (see Figure 22). For increases in the buffer layer thickness to 2 nm, the contact resistance increased (see Figure 22g). The conductivity improvement is explained by the de-pinning effect, as shown in Figure 22e, and the barrier-lowering effect due to the interface dipole (see Figure 22f). When the Ta_2_O_5_ thickness was greater than 2 nm, the contact resistance also increased as the tunneling probability decreased with the thicker buffer layer. Therefore, there is an optimal range of the buffer layer thickness for minimizing the contact resistance of the MoS_2_/Ta_2_O_5_/Ti structure. Many previous studies have neglected the bandgap or band offset between the insulator and semiconductor in their analysis. However, as discussed in Section 2, in relation to (8), these factors also affect the tunneling probability at the semiconductor/insulator interface. Therefore, they should be considered important parameters for optimizing the process.

Shen et al. recently discovered that a Bi semimetal layer sandwiched between the MoS_2_ and Au electrode can effectively lower the MIGS density and the contact resistance [59]. This is because the Bi Fermi level is slightly above the conduction band edge of MoS_2_, resulting in a negligible barrier height and a high potential for achieving good ohmic contacts. Moreover, the density functional theory (DFT) calculation shows that the *p*_z_ orbital of Bi resonates with the *p*_z_ and *d*_z_^2^ orbitals of MoS_2_, shifting the Fermi level of MoS_2_ into the conduction band and creating a degenerate state of MoS_2_ (see Figure 23a). In this situation, the MIGS are fully saturated with electrons, and as the valence band states decrease faster than the MIGS increase (see Figure 23b), this, in effect, reduces the MIGS level [59].

One of the challenges in fabricating MIS devices based on 2D materials is the difficulty of growing a uniform oxide layer on 2D materials due to the lack of evenly distributed surface dangling bonds [100,101]. A possible solution to this challenge is to use hexagonal boron nitride (h-BN) as an interlayer. Wang et al. [91] demonstrated that inserting one or two layers of h-BN between Ni and MoS_2_ can significantly reduce the Schottky barrier from 158 meV to 31 meV and, consequently, lower the contact resistance from 5.1 kΩ·mm to 1.8 kΩ·mm. Similarly, by inserting a h-BN layer between MoTe_2_ and Sc, a high-performance n-type MoTe_2_ FET was achieved [101]. One way to gain better control of the threshold voltage and reliability in CMOS technology is to use surface or dangling bond passivation. This technique can also be applied to 2D materials. Cho et al. demonstrated chemisorbed thiol molecules as a tunneling layer at the interface between MoS_2_ and metal [102]. Thiol molecules can eliminate interface states that result from sulfur vacancies and create additional tunneling channels for charge injection through field emission. This lowers the interface barrier and enables ohmic contacts.

## 5. 2D Contact with Silicon

The current 2D-material-based electron devices are still much larger than state-of-the-art CMOS technology. There is a lack of mass production and large-scale integration technology for these devices. Therefore, it is unlikely that 2D-material-based ICs will replace the mainstream silicon technology in the next ten years. However, it is highly possible that some 2D materials will be integrated with Si technology to overcome some of the limitations of CMOS devices and fabrication technology and to enhance the performance of silicon devices. For instance, using 2D materials instead of the expensive indium tin oxide (ITO) films in photonic devices could be a promising option for 2D materials/Si technology integration. Graphene–silicon solar cells have been extensively studied since the discovery of graphene [103,104,105]. However, some fundamental issues of the graphene/silicon interface are still not well understood. Some puzzling issues, such as the wide variations in Schottky barrier height and large fluctuations in the ideality factor (from ~1 to 30), were observed in this simple structure [41,96]. These wide ranges in parameter values suggest that some additional physical mechanisms in addition to the Schottky emission should be considered.

The interface between 2D material and silicon has a unique nature that is not common in the conventional silicon process and is also unknown to the 2D material community [41]. Figure 23 shows the different types of material interfaces [41]. Two-dimensional materials, such as graphene, have a much lower surface defect density than most conventional semiconductor materials. However, silicon has many dangling bonds on its surface. The van der Waals contact of the graphene/Si interface would leave a large number of unpassivated silicon dangling bonds. Figure 24a depicts a freshly etched <100> silicon surface full of silicon dangling bonds, also called P_b0_ centers or denoted as ≡Si. [106,107,108]. A P_b0_ center can capture both electrons and holes. It is an amphoteric defect center. Thus, it would contribute to the current conduction in both forward and reverse biases. In MOS transistors or Si technology, defects can be readily passivated with well-developed forming gas annealing. In the forming gas annealed Si surface or Si/SiO_2_ interface, the defects are passivated with a hydrogen- or water-related hydroxyl group (see Figure 24b). The defect density can be greatly reduced If the surface is oxidized (see Figure 24c) or bonded to other materials, such as metal (see Figure 24d). Figure 24e,f illustrate the graphene/Si contact. Graphene may be able to form covalent bonds, such as in Gr/SiC structures and edge contact, as mentioned in Section 4.1. For 2D van der Waals contacts, there is no strong physical bonding, either covalent or ionic, between the silicon and the 2D graphene layer. This is a distinct contact that cannot be found in conventional semiconductor contacts. Because the silicon dangling bonds are unpassivated, the number of defects is significantly higher than that of other interfaces. This results in significant Fermi-level pinning. However, as the hexagonal openings in the graphene lattice are so small that most of the atoms or molecules are too large to pass through them, the defects are physically isolated from other materials. This model was used to explain the unusual current–voltage characteristics of graphene/Si contacts [41].

Figure 25 shows the forward current–voltage characteristics of 2D material/Si Schottky junctions from different sources [41,100]. We can see that these I–V curves do not follow the expected Schottky behavior. The Schottky equations do not fit the current well, and the ideality factors from the fitting are very large and inconsistent. This may be due to the oxide layers or interface defects in the junctions [41]. These factors can affect the charge transport mechanisms, such as thermionic emission, thermionic field emission, or tunneling, and increase the junction current, especially at high bias voltages. Figure 26 illustrates how P_b0_ centers can contribute to the current’s conduction under both forward and reverse biases [41]. The main current under forward bias is caused by the thermionic emission of electrons from the silicon conduction band to the graphene conduction band via mechanism ①. Some of these electrons may be trapped by the P_b0_ centers, as shown in mechanism ②. The trapped electrons may be emitted into the conduction band of graphene via mechanism ③. These charge transport processes can result in abnormal values of the effective barrier height and ideality factor if one fits the I–V characteristics with the Schottky equation as provided in (7) or (8). Additional physical mechanisms in addition to the Schottky emission should be considered, which also implies possible techniques for improvements in the contact characteristics.

In summary, the interface between 2D materials and silicon is different than other interfaces. The contact behavior and the contact stability depend mainly on the surface properties of the silicon, especially the surface defects and how they are treated. The Schottky equation is not very accurate for describing the current flow in this case.

## 6. Concluding Remarks

As CMOS technology further shrinks to the nanoscale range in the coming technological nodes, aggressive contact scaling is indispensable. This scaling will lead to some poor contacts because of the more significant impact of the surface roughness, contact size reduction, and the nonscalability of film interfaces, as well as the use of undoped substrates. These factors lead to higher contact resistance and larger characteristic nonlinearity, which could hinder performance improvement in further-scaled CMOS devices and integrated circuits. On the other hand, although the emerging two-dimensional materials with smooth surfaces and dangling-bond-free nature have the potential to offer better material interfaces and form good electrical contacts, achieving signal transparent ohmic contacts is still a big challenge. The contacts between two-dimensional materials and semiconductors, as well as between two-dimensional materials and metals, usually have large Schottky barriers, and they do not follow the Schottky–Mott rule due to surface defects pinning the Fermi level and the van der Waals nature of the material interfaces.

In this review, we examined some of the theoretical background of metal/semiconductor contacts and highlight the validity of the Schottky equation. Then, we discussed the issues associated with reducing the contact size and the effects of a nanometer CMOS device’s structure and fabrication technique on ohmic contacts. We provided a systematic survey of the recent advancements and technological trends in contact engineering, such as phase and bandgap engineering, 2D/metal van der Waals contacts and hybrid contacts, junction doping technology, contacts with a buffer layer, for emerging 2D materials. This technological overview should offer new insight and solutions for overcoming contact scaling issues and facilitates the technological development for abridging the 2D materials with future CMOS technology.

## Figures and Tables

**Figure 1 nanomaterials-14-00386-f001:**
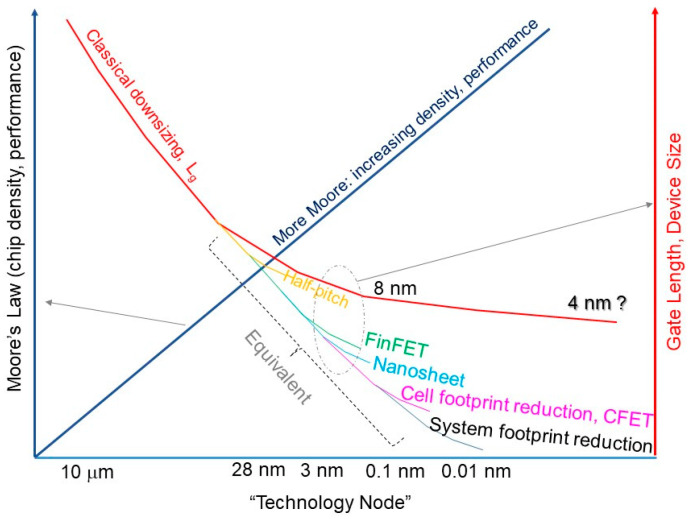
Illustration of more Moore strategies for downsizing effective technology nodes beyond the decananometer and subnanometer ranges.

**Figure 2 nanomaterials-14-00386-f002:**
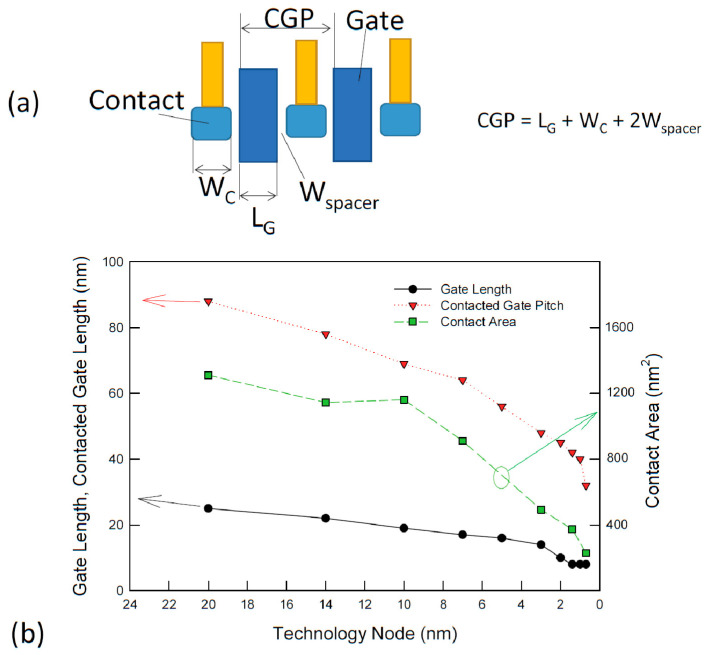
(**a**) Definition of contacted gate pitch (CGP) or contact poly pitch (CPP); (**b**) trends in gate length, contacted gate length, and contact area downsizing. Data taken from [5,12].

**Figure 3 nanomaterials-14-00386-f003:**
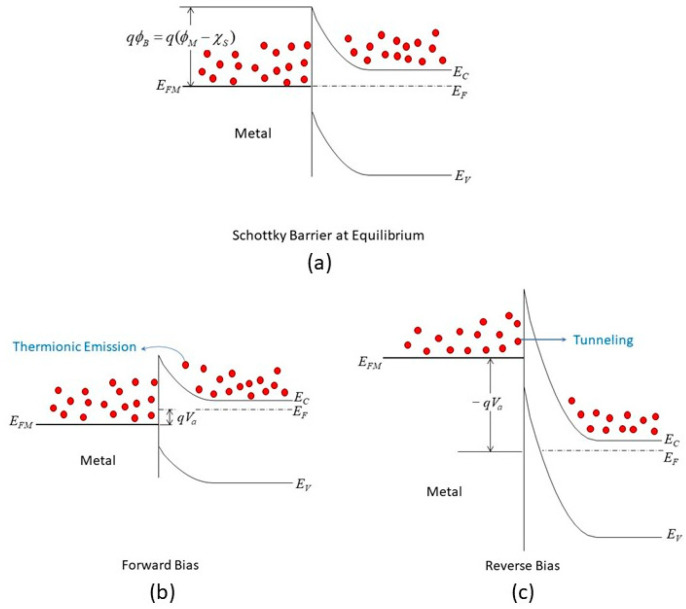
Metal/n-type semiconductor Schottky contact: (**a**) equilibrium; (**b**) forward bias; (**c**) reverse bias.

**Figure 4 nanomaterials-14-00386-f004:**
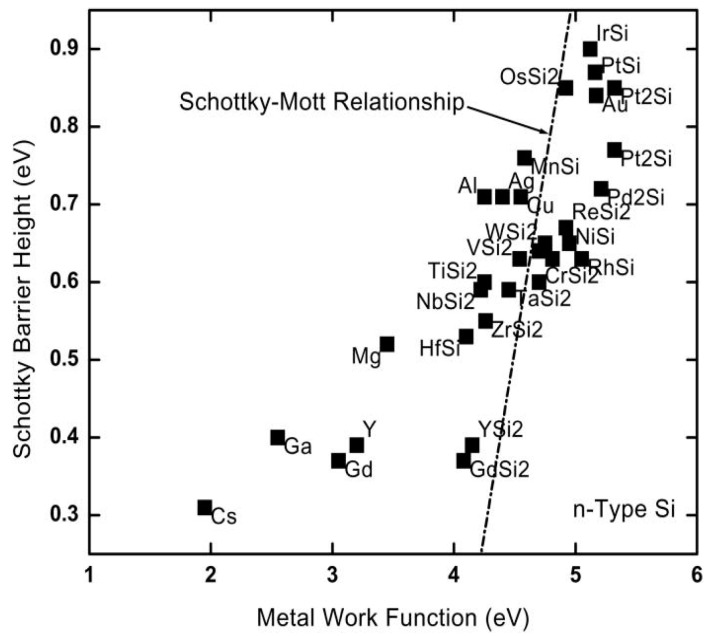
Plot of the experimental Schottky barrier heights for different metals on n-type Si taken from various sources. The straight line indicates the Schottky–Mott rule [36]. © 1993 American Vacuum Society. Reproduced with permission.

**Figure 5 nanomaterials-14-00386-f005:**
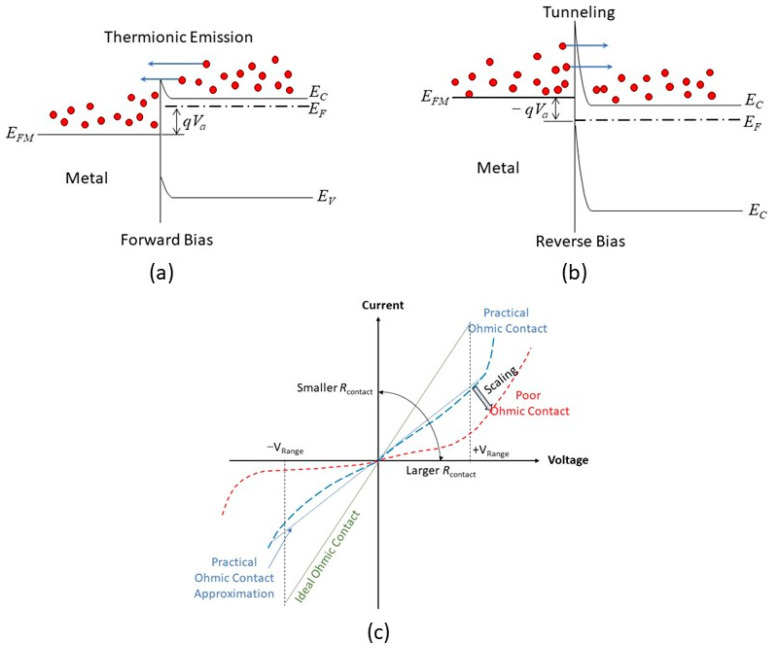
Ohmic contact and current-conducting mechanisms: (**a**) forward bias; (**b**) reverse bias. (**c**) Illustration of the current–voltage characteristics of an ideal ohmic contact, practical ohmic contact, and poor ohmic contact that may result from downsizing.

**Figure 6 nanomaterials-14-00386-f006:**
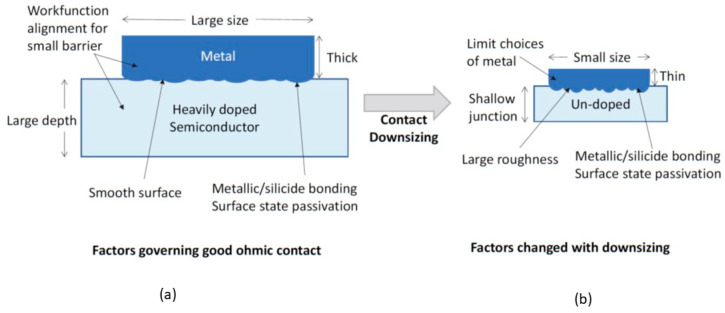
(**a**) Factors governing the ohmic contact; (**b**) issues encountered when the device technology scales down to the nanometer range.

**Figure 7 nanomaterials-14-00386-f007:**
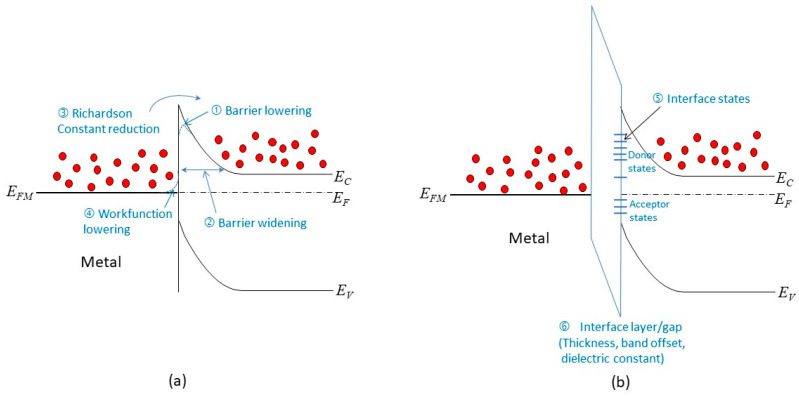
(**a**) Possible effects resulting from a scaled metal/semiconductor contact; (**b**) issues associated with a contact with an interlayer.

**Figure 8 nanomaterials-14-00386-f008:**
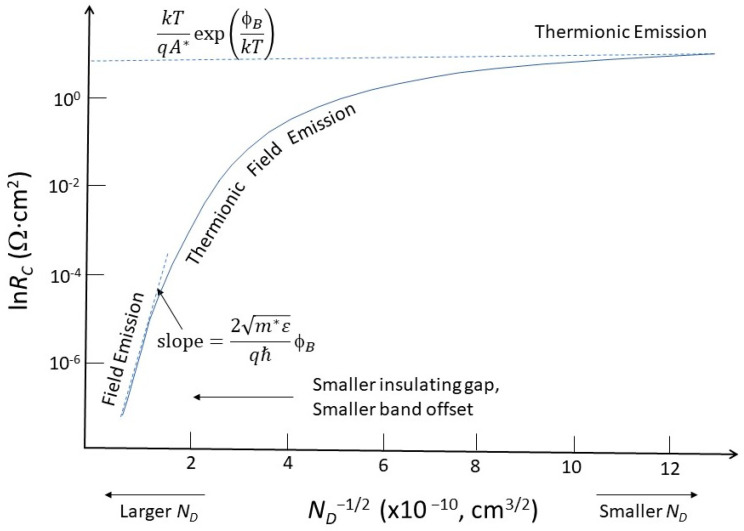
Illustration of doping concentration dependence of contact resistance involving different current conduction mechanisms.

**Figure 9 nanomaterials-14-00386-f009:**
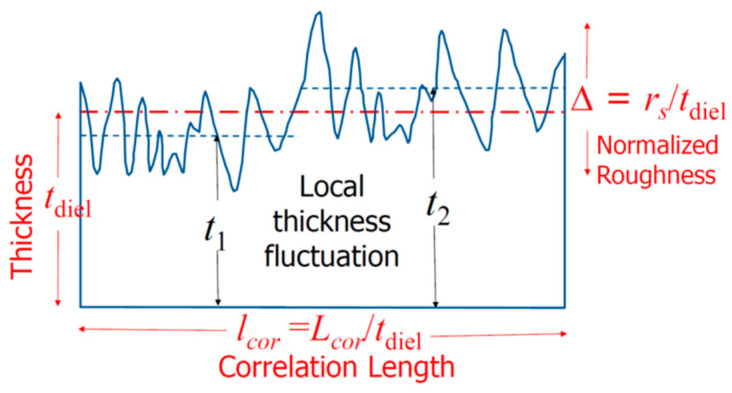
Definitions of the key roughness parameters and an illustration of the local averaged thickness variation for small-sized devices as a function of the applied field [50].

**Figure 10 nanomaterials-14-00386-f010:**
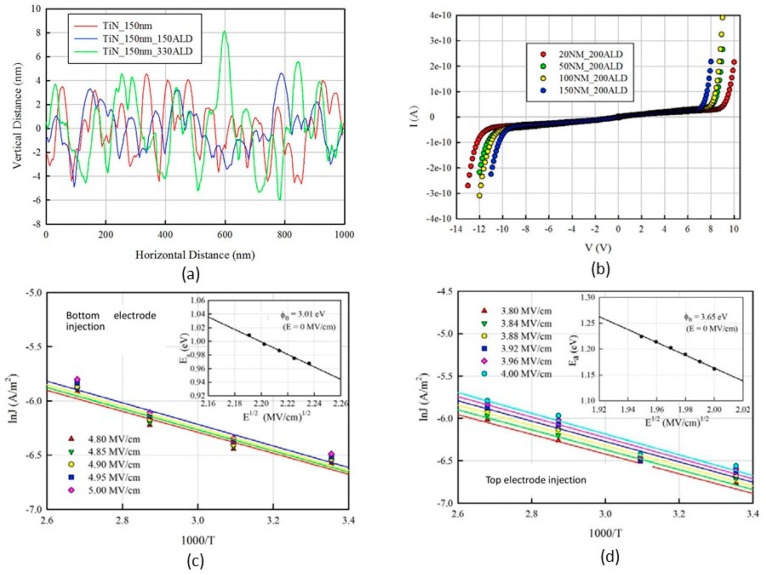
(**a**) Comparison of the surface thickness fluctuations of a nude TiN sample, TiN with a 20 nm thick Al_2_O_3_ deposited by 150 °C ALD, and TiN with a 20 nm thick Al_2_O_3_ deposited by 330 °C ALD; (**b**) typical current–voltage characteristics of TiN/Al_2_O_3_/TiN MIM capacitors showing the asymmetric forward and reverse leakage currents and bottom electrode dependencies; (**c**) extraction of the bottom barrier height from the temperature-dependent forward I–V characteristics; (**d**) top barrier from the reverse characteristics [52]. © 2018 Elsevier. Reproduced with permission.

**Figure 11 nanomaterials-14-00386-f011:**
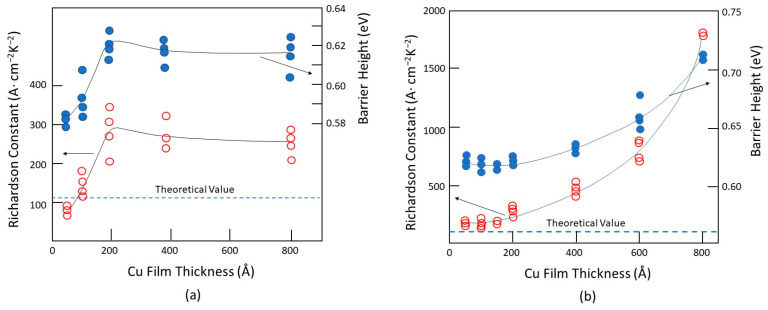
The metal thickness and deposition process are dependent on the Schottky barrier height and Richardson constant: (**a**) Cu-Si contacts by evaporation; (**b**) sputtered Cu-Si contacts prepared by sputtering. Redrawn based on [46].

**Figure 12 nanomaterials-14-00386-f012:**
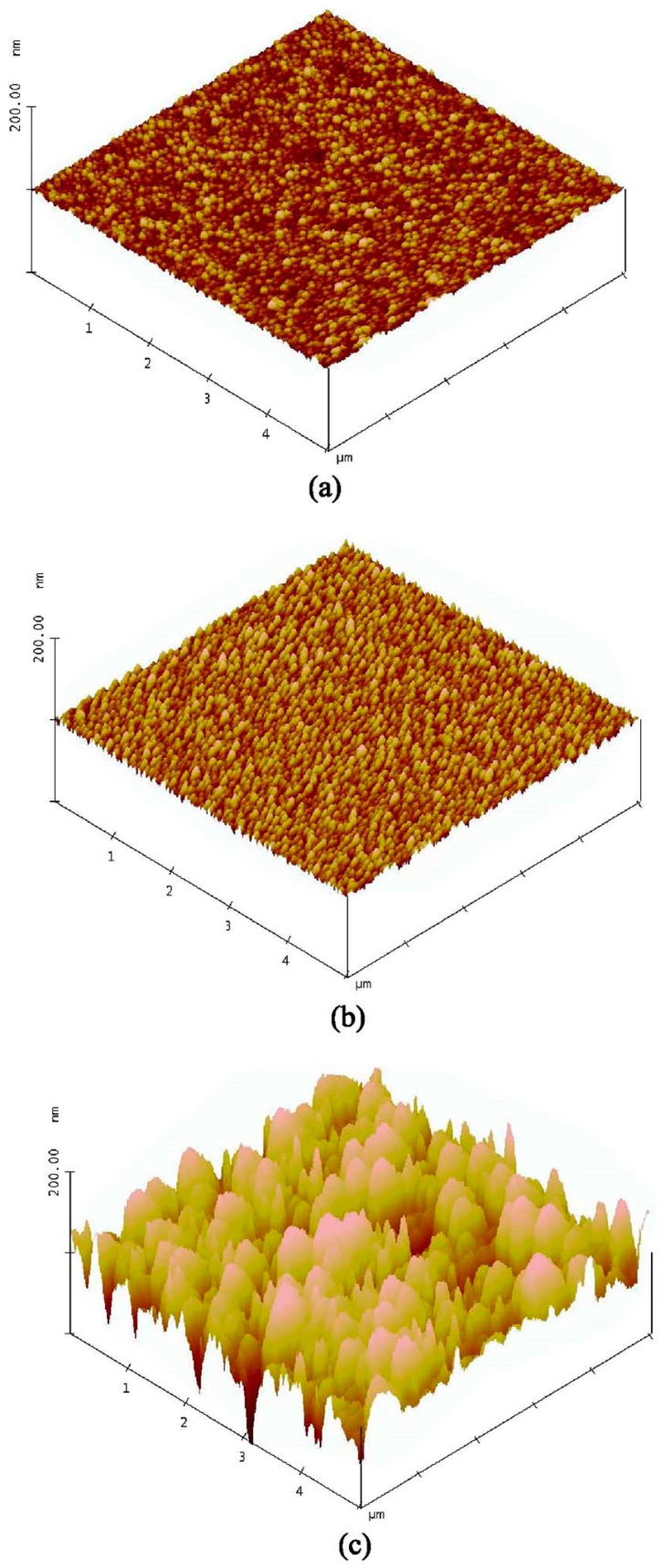
AFM picture of atomic layer deposition of copper on Ru substrates: (**a**) 5.1 nm with an RMS roughness of 1.5 nm; (**b**) 3.4 nm thick with an RMS roughness of 3.4 nm; (**c**) 4.7 nm thick with an RMS roughness of 21 nm [58]. © 2016 Electrochemical Society. Reproduced with permission.

**Figure 13 nanomaterials-14-00386-f013:**
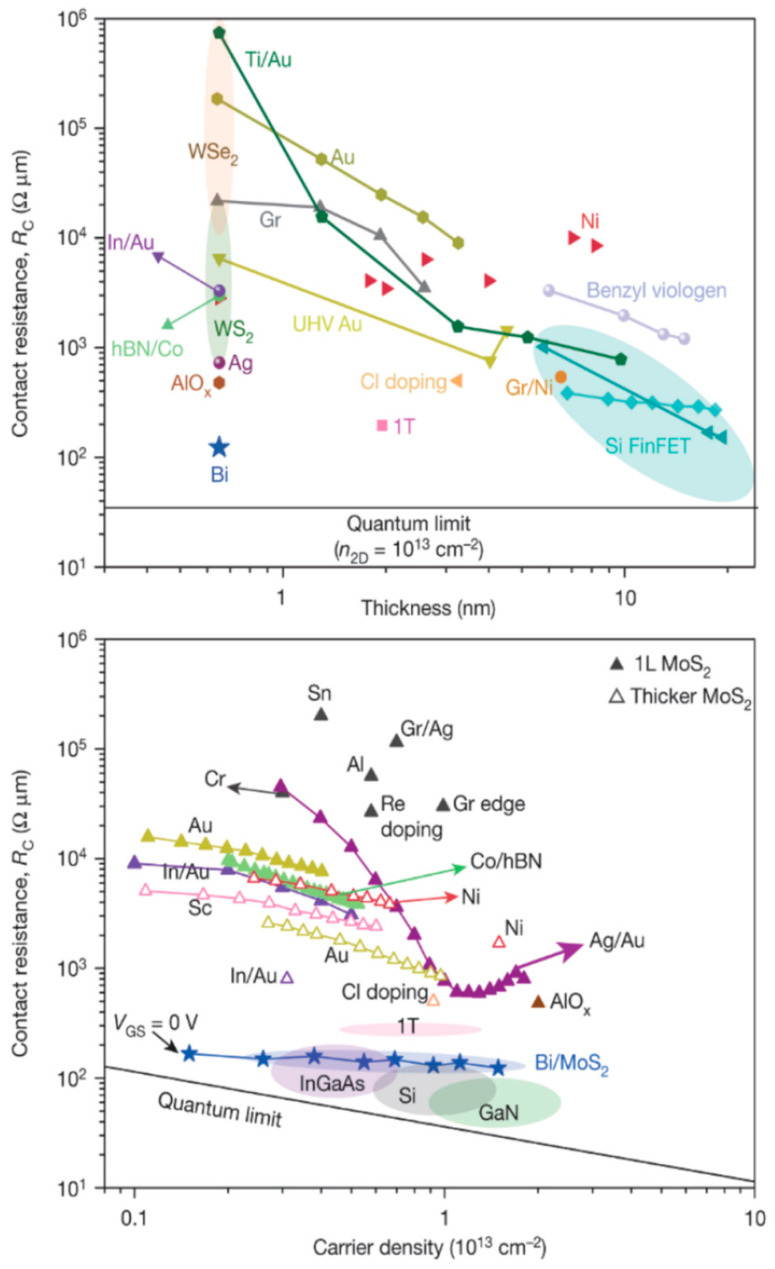
(**Top**) Plot of the contact resistance as a function of the film thickness or Si fin thickness in the case of FinFET; (**Bottom**) contact resistance as a function of the carrier density. Data are from various sources [59]. © 2021 Springer Nature. Reproduced with permission.

**Figure 14 nanomaterials-14-00386-f014:**
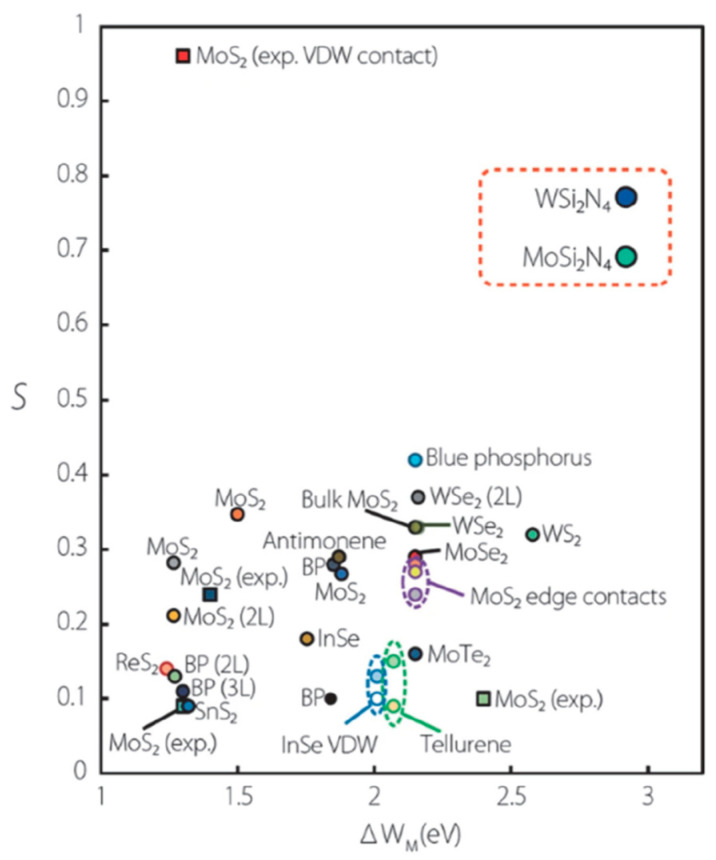
Plot of the pinning parameter S versus the metal workfunction range (ΔW_M_) of various 2D materials from various sources. The labels “2L” and “3L” in brackets denote a bilayer and trilayer, respectively [62]. © 2021 Springer Nature.

**Figure 15 nanomaterials-14-00386-f015:**
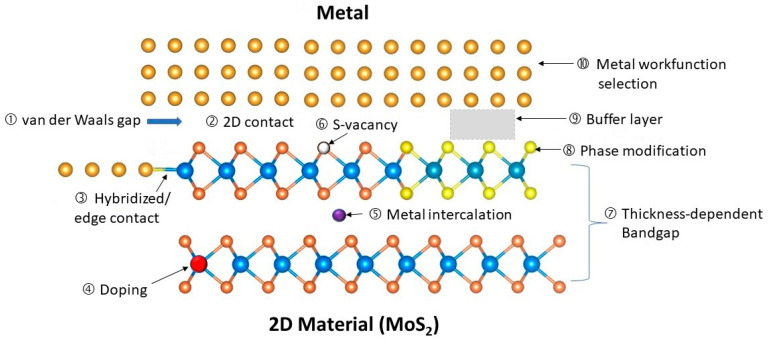
Illustration of the issues and features associated with metal–2D material (MoS_2_, as an example) contacts.

**Figure 16 nanomaterials-14-00386-f016:**
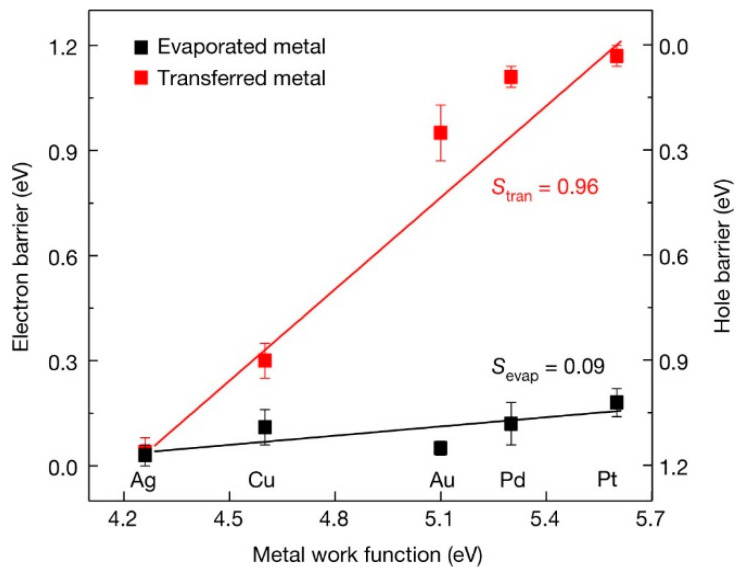
Comparison of metal/MoS_2_ Schottky barrier contacts produced by thermal evaporation and transferred from predeposition on atomically flat silicon surface [63]. © 2018 Springer Nature. Reproduced with permission.

**Figure 17 nanomaterials-14-00386-f017:**
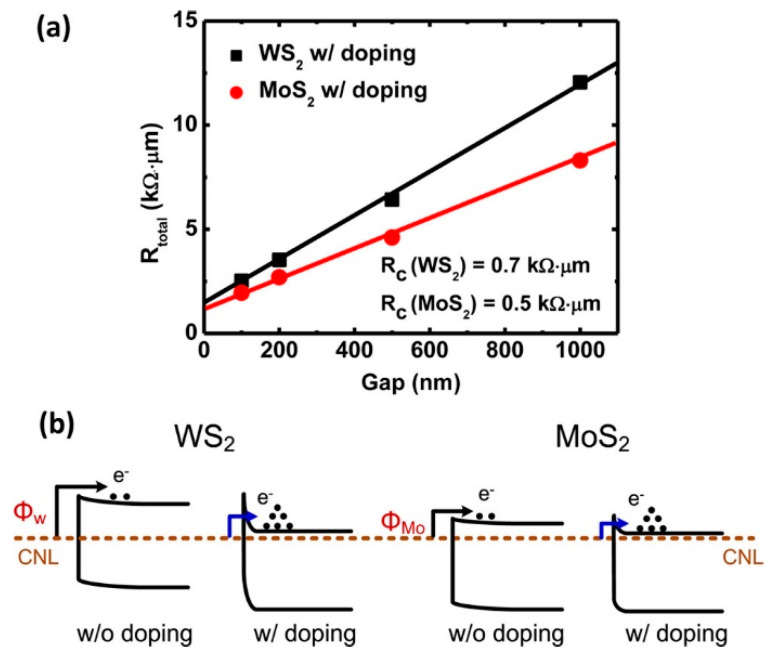
Comparison of the Cl doping’s effect on the contact resistance of WS_2_ and MoS_2_ contacts: (**a**) plot of the contact resistances as a function of the gap spacing between two electrodes; (**b**) schematic band diagram of the doping effects of the studied contacts. The Fermi level was pinned close to the charge neutrality level (CNL) before doping, and the Fermi level moved upward in the heavily doped 2D materials [74]. © 2014 American Chemical Society. Reproduced with permission.

**Figure 18 nanomaterials-14-00386-f018:**
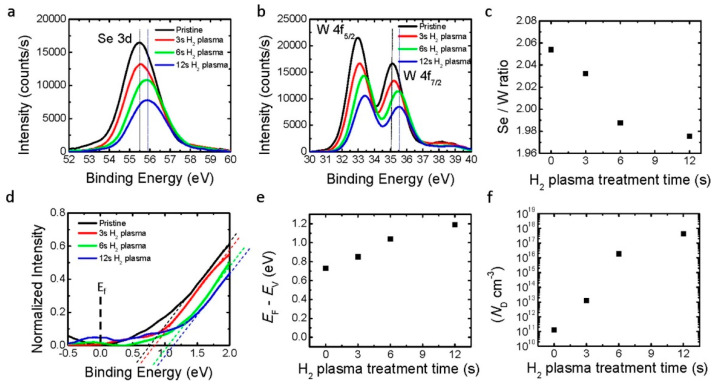
Various characteristics produced by hydrogen plasma treatment of WSe_2_ 2D materials: (**a**,**b**) evidence of Se reduction with a reduced Se 3d XPS peak intensity and energy shift of W 4f spectra; (**c**) Se/W ratio as a function of the plasma treatment time; (**d**) normalized valence band spectra; (**e**,**f**) change in the Fermi level and the electron doping concentration as a function of the plasma treatment’s duration [75]. © 2016 American Chemical Society. Reproduced with permission.

**Figure 19 nanomaterials-14-00386-f019:**
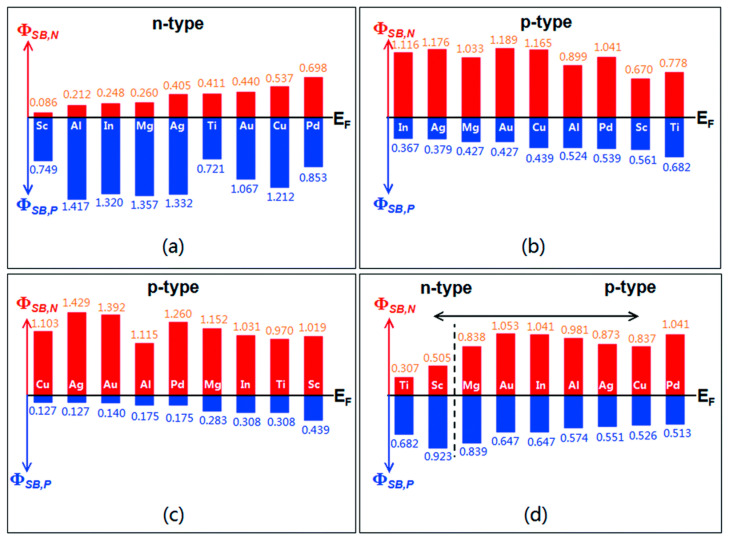
Theoretical Schottky barrier height for various metals in different contacts with monolayer MoS2: (**a**) top contact; (**b**) edge contact through connecting Mo and S atoms at an armchair termination; (**c**,**d**) edge contact through connecting Mo and S atoms at a zigzag termination, respectively [81]. © 2019 Royal Society of Chemistry. Reproduced with permission.

**Figure 20 nanomaterials-14-00386-f020:**
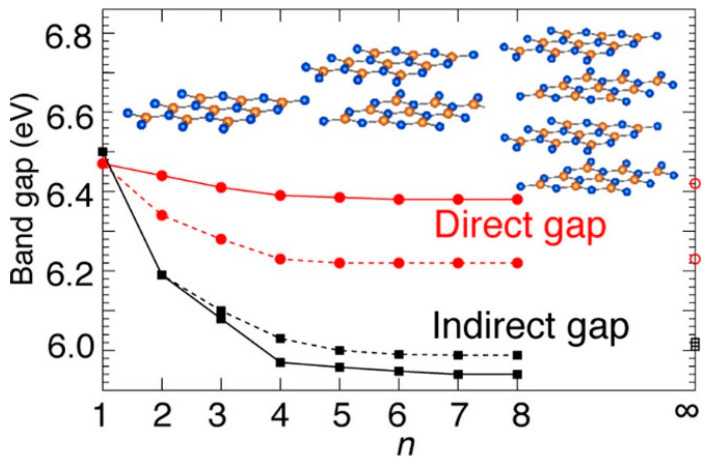
Variation in the bandgap and changes in the gap types of hexagonal boron nitride with different monolayers calculated from the first principle [82]. © 2018 American Chemical Society. Reproduced with permission.

**Figure 21 nanomaterials-14-00386-f021:**
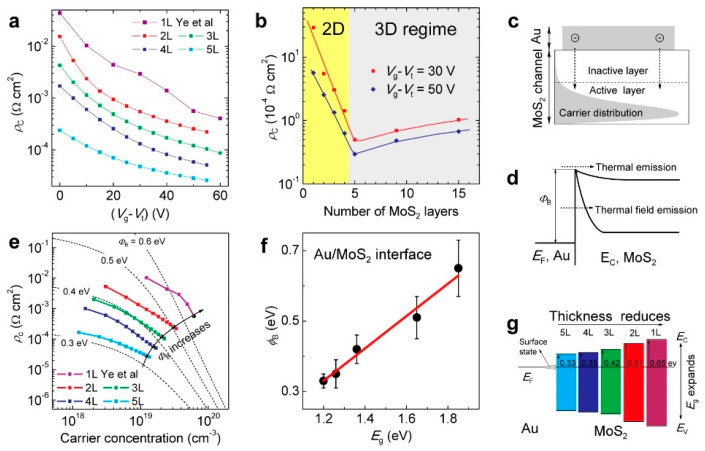
Demonstration of the dependence of the specific contact resistivity on the MoS_2_ thickness: (**a**) gate bias dependence; (**b**) thickness dependence; (**c**) carrier distribution and injection path; (**d**) carrier injection mechanisms; (**e**) experimental data indicating the change in the barrier height; (**f**) plot of the barrier height as a function of the bandgap; (**g**) illustration of the energy-level alignment at Au/MoS_2_ interfaces for different MoS_2_ thicknesses [83]. © 2014 American Chemical Society. Reproduced with permission.

**Figure 22 nanomaterials-14-00386-f022:**
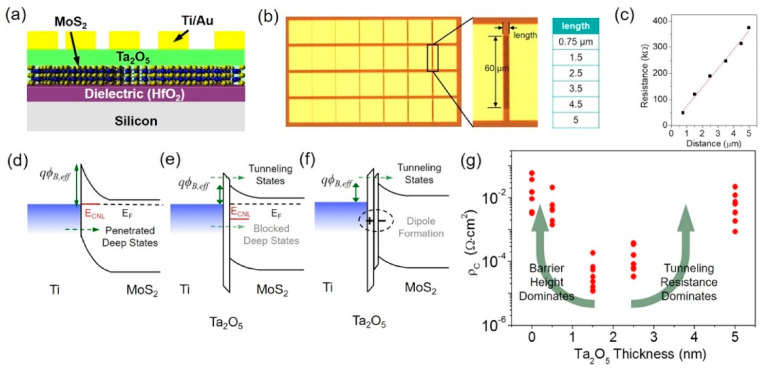
Effects of the Ta_2_O_5_ buffer layer on the contact characteristics of Au-Ti/MoS_2_ contacts: (**a**,**b**) device structure, layout, and scales; (**c**) resistance value as a function of the contact separation; (**d**–**f**) explanation of the Fermi-level pinning and the impact of the Ta_2_O_5_ buffer layer; (**g**) plot of the measured specific contact resistivity, ρ_c_, as a function of the Ta_2_O_5_ dielectric thickness [99]. © 2016 American Chemical Society. Reproduced with permission.

**Figure 23 nanomaterials-14-00386-f023:**
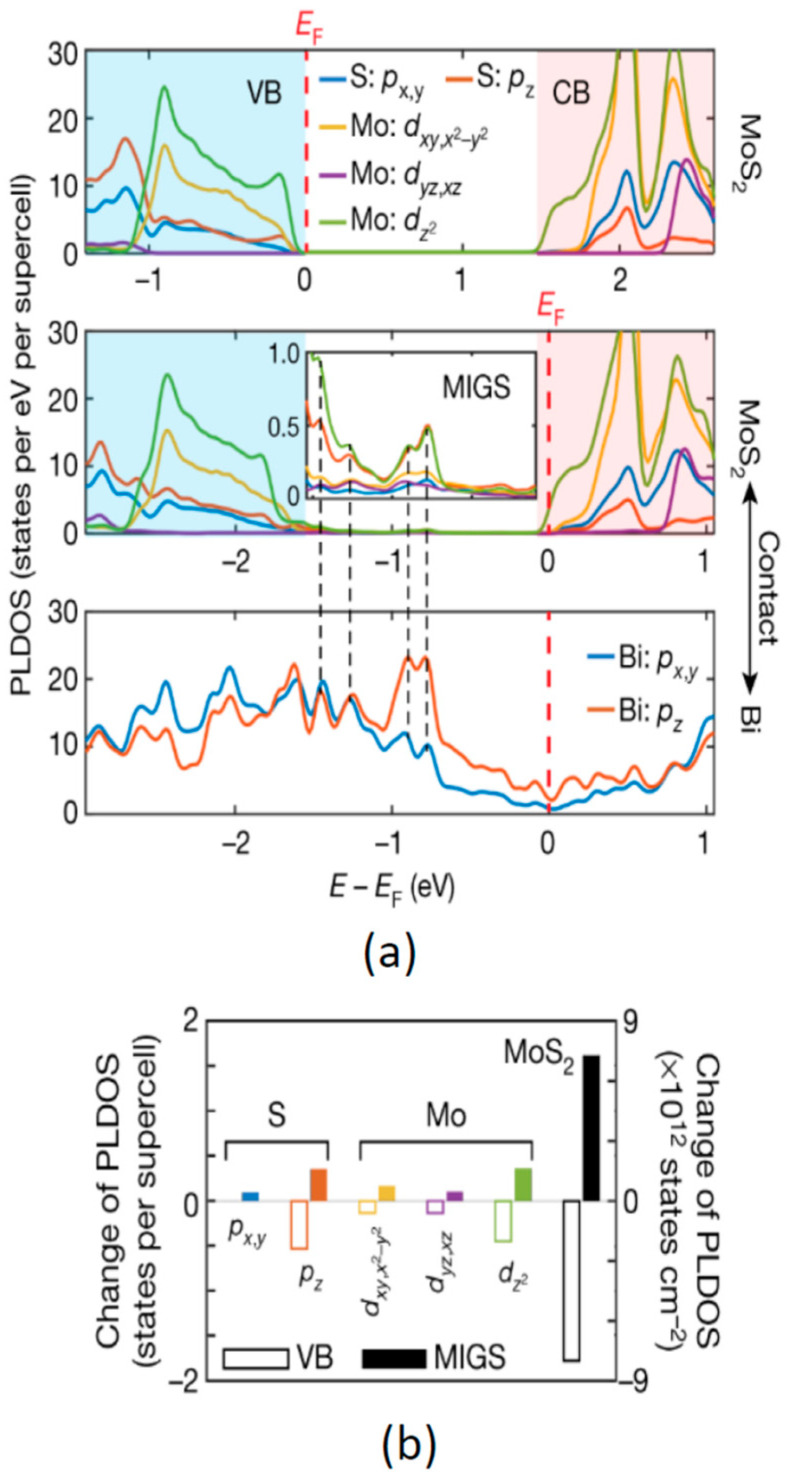
(**a**) Density functional theory calculation results of the projected local density of states (PLDOS) of the MoS_2_ (**top**), MoS_2_ in contact with Bi (**middle**), Bi (**bottom**); (**b**) change in PLDOS of different orbitals in the valence band and MIGS region [59]. © 2021 Springer Nature. Reproduced with permission.

**Figure 24 nanomaterials-14-00386-f024:**
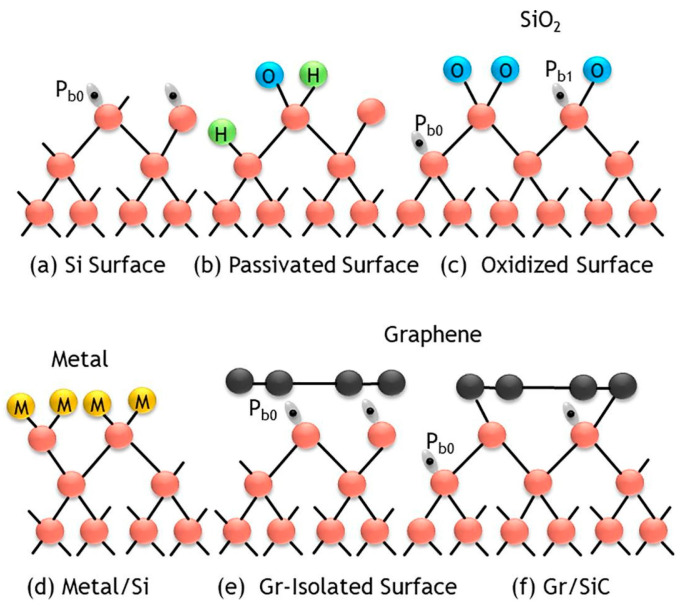
Six different surfaces/contacts with Si: (**a**) bare silicon surface and surface dangling bonds or P_b0_ centers; (**b**) P_b0_ centers passivated with hydrogen and hydroxyl; (**c**) silicon covered by silicon oxide; (**d**) metal/Si contact; (**e**) graphene/Si contact; (**f**) Si/carbon covalent contact [41]. © 2021 Elsevier. Reproduced with permission.

**Figure 25 nanomaterials-14-00386-f025:**
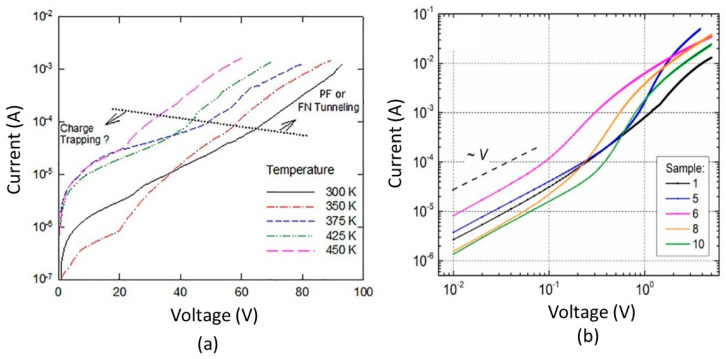
Abnormal current–voltage characteristics of a graphene/n-type Si Schottky diode taken from Ref. [41] (**a**) and Ref. [109] (**b**). © 2021, 2022, Elsevier. Reproduced with permission.

**Figure 26 nanomaterials-14-00386-f026:**
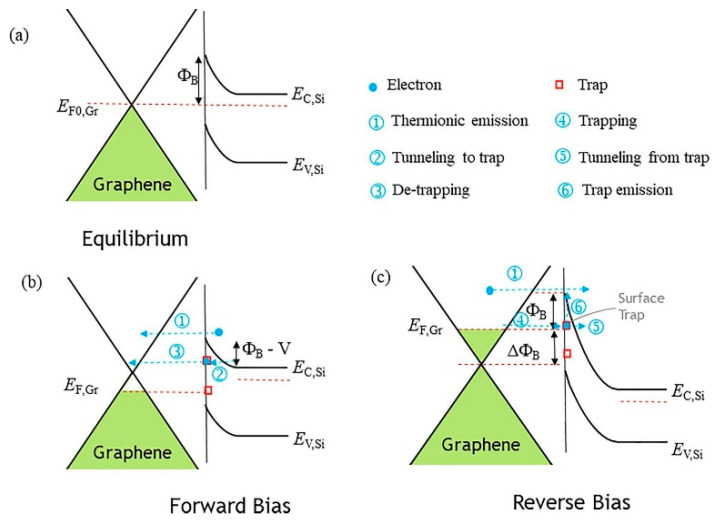
The involvement of silicon P_b0_ centers in the current conduction of graphene/Si junctions: (**a**) equilibrium; (**b**) forward bias; (**c**) reverse bias [41]. © 2021 Elsevier. Reproduced with permission.

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
