# Peer review of "Contacts at the Nanoscale and for Nanomaterials"

_nanomaterials, 2024, doi:10.3390/nano14040386_

Round 1

Reviewer 1 Report

Comments and Suggestions for Authors

In this review, the authors discuss a significant challenge in nano CMOS (Complementary Metal-Oxide-Semiconductor) technology known as contact scaling. As technology shrinks to the nanometer range, issues related to surface roughness, contact size, film thicknesses, and undoped substrate become more problematic. These factors contribute to increased contact resistance and nonlinearity in current-voltage characteristics. The review explores how these challenges may limit the benefits of further downsizing CMOS devices.

I believe that this review is suitable for publication, also in view of the extensive research conducted on this topic over the years. I am confident that by addressing the following points, the authors can potentially improve the clarity and soundness of their arguments and contribute to the overall quality of the article.

1)     Line 76: the sentences referring to each material (MoS2, MoTe2, WSe2) should have some supporting reference. I suggest these articles:

·       Low-temperature MoS2 growth on CMOS wafers. Nature Nanotechnology, 1-2 (2023).

·       Electron irradiation of metal contacts in monolayer MoS2 field-effect transistors. ACS Applied Materials & Interfaces, 12(36), 40532-40540 (2020).

·       Temperature-dependent photoconductivity in two-dimensional MoS2 transistors. Materials Today Nano, 24, 100382 (2023).

·       Optoelectronic memory in 2D MoS2 field effect transistor. Journal of Physics and Chemistry of Solids, 179, 111406 (2023).

·       Homogeneous 2D MoTe2 CMOS Inverters and p–n Junctions Formed by Laser‐Irradiation‐Induced p‐Type Doping. Small, 16(30), 2001428.4 (2020).

·       Homogeneous 2D MoTe2 p–n junctions and CMOS inverters formed by atomic‐layer‐deposition‐induced doping. Advanced Materials, 29(30), 1701798 (2017).

·       Improvements in 2D p-type WSe2 transistors towards ultimate CMOS scaling. Scientific reports, 13(1), 3304 (2023).

·       High-performance WSe2 complementary metal oxide semiconductor technology and integrated circuits. Nano letters, 15(8), 4928-4934 (2015).

2)     Line 134: the authors should discuss the limitations and exceptions of the Schottky-Mott rule in predicting the barrier height due to pinning at the Fermi level.

3)     Line 186: the authors should add reference regards the Ideality Factor (n) in Schottky Diodes.

4)     Line 193-195: the authors should elaborate on the physical or technical meaning of the ideality factor in Schottky diodes.

5)     Line 323: the section mentions experimental results suggesting that thinner metal films have lower work functions. Request the authors to cite specific references or provide additional details on these experimental findings to strengthen the argument and provide readers with a basis for further exploration.

6)     Line 331: the authors discuss the use of an interface layer for passivation. The authors should elaborate on the types of materials considered for the interface layer.

7)     Line 441: the authors should discuss the implications of large grain size and high surface roughness in ALD-deposited films.

8)     Line 480: the authors should add the reference “A current–voltage model for double Schottky barrier devices. Advanced Electronic Materials, 7(2), 2000979. (2021)”.

9)     The authors should expand the statement in line 699. They should discuss the specific benefits and distinctions, highlighting how this feature can be exploited to improve device performance in 2D material-based technologies.

10) Line 841: the authors should clarify the contribution of Pb0 centers to current conduction in the presence of both forward and reverse polarizations, as shown in Figure 25.

11) Line 900:  in the sentence “In this review, we have reviewed some theoretical background of the metal/semiconductor contacts and highlight the validity of Schottky equation.” the authors should replace “reviewed” with “examined”.

Author Response

All comments from the reviewer have been adopted in the revision. Point-to-point responses to the reviewer's comments are given in the attached file. 

Reviewer 2 Report

Comments and Suggestions for Authors

In this review paper, the authors mainly talked about the nanoscale contact and the contact for nanomaterial.

However, this review needs a major revision due to the following reasons:

1.  Section 2 includes too much details about the Schottky Junction. The author may want to make it more concise since this knowledge can be found in most semiconductor textbook

2.  At the beginning of Section 3, the authors list six effects while scaling down the contact. However, only the doping and the surface roughness are discussed later in this section.  The authors should consider including some recent works on Si contact improvement, such as fermi level unpinning. 

3. there are two figure 12. The first one (the AFM picture) is pointless and should be removed.

4. "contact for 2D material" may be a better title for section 4 since it mainly talks about 2D material

5. the citation of Figure 12 (section 4) is incorrect. This figure is directly snipped from ref[51].  The bottom plot still says "this work" which is inappropriate here. 

In addition, many figures are snipped from the original paper, which makes them blurry and hard to read.

6.  Line 474, "Because only a few monolayers are thick" not sure what this means

7.  Line 817, cannot understand the title. If the authors want to say 2D contact with Silicon, maybe this part can be combined with Section 3

8. the authors may want to include the key figure of ref[51] and then discuss how it mitigates the effect of MIGS.

Author Response

(The authors gave the same response as above.)

Reviewer 3 Report

Comments and Suggestions for Authors

Dear Authors, the review submission Contacts in Nanoscale and Contacts for Nanomaterials, Manuscript ID: nanomaterials-2861983, has some weak issues that must be revised appropriately.

Please find below some, of the most significant comments:

1.      In the Abstract section, the shortcuts or abbreviations should be omitted. Authors must present their short forms when first presented in the body text of the manuscript.

2.      Further, the Abstract section must be rewritten which contains many sentences starting with ‘we’.

3.      This section (Abstract) can be also clarified with the main direction of the study.

4.      Each of the cited items in the Introduction section must be presented separately, not [1-5]. Some of them are referenced later but must be highlighted with their novelty separately.

5.      In the subsection 2.1. Schottky Equation each of the cited equations must be referenced to the primary sources. As the proposed manuscript is a review, the Authors must refer to each of the studies included.

6.      Still with the 2.1 subsections, an advantage of the Schottky method over others presented (e.g. by Nordheim, Frenkel, Fowler or Mott) must be emphasized. If not, each of the studies must be included with its pros and cons.

7.      The limitations of the Ohmic method (section 3) must be provided. Authors must put a more comprehensive critical review. In its current form, it has more including advantages than any potential future aspects.

8.      In some cases, Authors use mental shortcuts, sentences are not clear and the reader is lost. For example: (1) Furthermore, due to the high oxygen content in the sputted films, the electronegativity of oxygen has a stronger binding with the electrons, and it makes the thermionic emission of electrons more difficult. (2) The challenges come from the requirement of even lower contact resistance to be achieved with the aggressively shrunk contact area and film thickness. (3) Most of the reported 2D material-based devices currently are much bigger than the 471 current CMOS technology, even though 2D materials are often linked to nanodevice and 472 nanotechnology in the literature. Please, try to justify all of the issues presented.

9.      In subsection 4.3, some more critical words must be included, for instance: It should be noted that Fang et al. reported that the top contact Schottky barrier height of monolayer MoS2 could not be accurately determined for Sc and Ti because of their strong interaction with MoS2 [72]. Limitations of the Schottky method are unknown or must be highlighted.

10.  Referencing the sentence: However, it is not sufficient to explain the contact resistivity (see Fig.20(e)) unless the barrier height increase is considered. Why? Further sentences do not justify those words and put them with assumptions.

11.  With the words: The origins of the FLP effect at the 2D material/metal interface should be due to several causes. I cannot find if they were listed. Or maybe origin not origins?

12.  The sentence: Similar to Fermi level pinning in high-defect Si/metal oxide interface, the intrinsic defects or impurities, such as sulfur vacancies or surface metal-like impurities on TMD materials, can act as electron donors [81]. must be expanded and include more relevant issues.

13.  Considering the sentences: A buffer layer can reduce the effects of metal-induced gap states (MIGS) by creating a matching layer between the 2D material and the metal. This idea is inspired by previous work on high-performance metal-insulator-semiconductor (MIS) diodes. The Authors should provide some details on the diodes' performance. In the current form, information lacks details and improvements.

14.  The problem formulated as The main current under forward bias is caused by the thermionic emission of electrons from the silicon conduction band to the graphene conduction band via mechanism… should be expanded and concluded. Many crucial information is lacking in the presented sentences.

15.  In Conclusion, the general advantages of the review presented must be emphasized by others. Currently, it is difficult to restrict, which are the most encouraging future prospects in the area of study.

16.  There are many abbreviations that the Nomenclature is required.

Generally, the proposed review manuscript is interesting, providing some crucial information on the topic addressed, however, includes some weaknesses that must be improved appropriately.

Therefore, in its current form, the manuscript is not suitable for publication in a quality journal as the Nanomaterials is, requiring proper improvements with a major revision, if allowed by a handling Editor.

Author Response

Point-to-point responses to the reviewer's comments are given in the attached file. 

Round 2

Reviewer 2 Report

Comments and Suggestions for Authors

The section number of Concluding Remarks should be 5 instead of 6. Other than that, this review is good to go.

Reviewer 3 Report

Comments and Suggestions for Authors

The responses were raised with their minimalist actions but the manuscript can be now considered for publication in its current, revised form.